# 3,4-Dichlorophenylacetic acid acts as an auxin analog and induces beneficial effects in various crops

Chao Tan[1,4], Suxin Li[1,4], Jia Song[1], Xianfu Zheng[2], Hao Zheng[2], Weichang Xu[2], Cui Wan[2], Tan Zhang[1], Qiang Bian [3✉] & Shuzhen Men [1✉]

Auxins and their analogs are widely used to promote root growth, flower and fruit development, and yield in crops. The action characteristics and application scope of various auxins are different. To overcome the limitations of existing auxins, expand the scope of applications, and reduce side effects, it is necessary to screen new auxin analogs. Here, we identified 3,4-dichlorophenylacetic acid (Dcaa) as having auxin-like activity and acting through the auxin signaling pathway in plants. At the physiological level, Dcaa promotes the elongation of oat coleoptile segments, the generation of adventitious roots, and the growth of crop roots. At the molecular level, Dcaa induces the expression of auxin-responsive genes and acts through auxin receptors. Molecular docking results showed that Dcaa can bind to auxin receptors, among which TIR1 has the highest binding activity. Application of Dcaa at the root tip of the *DR5:GUS* auxin-responsive reporter induces GUS expression in the root hair zone, which requires the PIN2 auxin efflux carrier. Dcaa also inhibits the endocytosis of PIN proteins like other auxins. These results provide a basis for the application of Dcaa in agricultural practices.

[1] Tianjin Key Laboratory of Protein Sciences, Department of Plant Biology and Ecology, College of Life Sciences, Nankai University, 300071 Tianjin, China. [2] Zhengzhou ZhengShi Chemical Co., Ltd, 450000 Zhengzhou, China. [3] National Pesticide Engineering Research Center (Tianjin), College of Chemistry, Nankai University, 300071 Tianjin, China. [4] These authors contributed equally: Chao Tan, Suxin Li. ✉email: bianqiang@nankai.edu.cn; shuzhenmen@nankai.edu.cn

The phytohormone auxin is essential for regulating plant growth and development and response to environmental signals[1–3]. Auxin controls organogenesis, pattern formation, and tropic growth through accumulating asymmetrically in tissues and organs[1,4–7]. Membrane-localized proteins including AUXIN RESISTANT1/LIKE AUX1 (AUX1/LAX) influx transporters, PIN-FORMED (PIN) efflux carriers, and ATP-BINDING CASSETTE subfamily B (ABCB) transporters mediate directed auxin flow and polar distribution[8,9]. PIN proteins display polar localization on the plasma membrane, which is established and maintained by endocytosis and transcytosis[10,11]. Auxin promotes PIN protein localization on the plasma membrane by inhibiting their endocytosis[12,13]. In the nucleus, auxin binds to the TRANSPORT INHIBITOR RESPONSE 1/AUXIN SIGNALING F-BOX (TIR1/AFB) receptors and promotes the degradation of the AUXIN/INDOLE-3-ACETIC ACID (AUX/IAA) transcriptional repressors by the 26S proteasome[2,14–16]. This removes the repression of AUX/IAAs on the AUXIN RESPONSE FACTOR (ARF) transcription factors, which regulate the expression of auxin-responsive genes[17].

Indole-3-acetic acid (IAA) is the major endogenous auxin in many plants and is the most studied. IAA is mainly synthesized from tryptophan (Trp) through a two-step reaction catalyzed by Trp-aminotransferase and YUCCA (YUC) flavin monooxygenase, respectively[18–23]. Besides IAA, plants synthesize three other endogenous auxins, indole-3-butyric acid (IBA), phenylacetic acid (PAA), and 4-chloroindole-3-acetic acid (4-Cl-IAA)[24–26]. IBA probably serves as an IAA precursor and storage form to regulate dynamic IAA levels in response to developmental and environmental signals[24,27,28]. PAA is widely present in plant species analyzed, such as oats, barley, rice, maize, wheat, tobacco, Arabidopsis, the moss *Physcomitrella patens*, and the liverwort *Marchantia polymorpha* and, its level is higher than that of IAA[25,29,30]. PAA can promote the interaction of the TIR1/AFB receptors with the AUX/IAA repressors, but its auxin activity is lower than that of IAA[30]. Unlike IAA, PAA is not transported in a polar manner[30–32]. 4-Cl-IAA is found in the seeds of several legume plants and has higher auxin activity than IAA[24,33,34]. 4-Cl-IAA is a substrate for the auxin influx and efflux carriers[35,36].

Synthetic auxin analogs such as 1-naphthaleneacetic acid (NAA) and 2,4-dichlorophenoxyacetic acid (2,4-D) are designed to mimic endogenous auxins and have been widely used in auxin research, plant tissue culture, and agriculture[37]. In agriculture, auxin analogs are used as herbicides and plant growth regulators (PGRs)[37–39]. Auxinic herbicides are generally selective against dicot plants and thus can be used to control broadleaf weeds in different crops[40]. 2,4-D and dicamba (3,6-dichloro-2-methoxybenzonic acid) are the two most commonly used auxinic herbicides worldwide[40]. Synthetic auxins used as PGRs include NAA, 2,4-D, 2-naphthoxyacetic acid, 1-naphthylacetamide, p-chlorophenoxyacetic acid, etc. The action characteristics and application scope of these synthetic auxins are different. For example, NAA has strong biological activity and is mainly used in micropropagation and *ex vitro* rooting[41]. It is also used in horticultural cultivation to promote flowering and increase fruit set and fruit size[42,43]. However, cucurbit crops are sensitive to NAA and are prone to a toxic effect. 2,4-D is used to promote fruit set, prevent preharvest fruit drop, and maintain postharvest fruit quality[44,45]. However, the use of 2,4-D is limited due to environmental and human health concerns[45]. Therefore, new auxin-like compounds must be discovered to overcome these limitations. Based on the molecular structures of those existing auxinic compounds, we have designed more than 2,000 chemicals. From them, 82 compounds were screened out for further physiological activity tests, among which the No. 066 compound (3,4-dichlorophenylacetic acid, Dcaa) was the most promising. Dcaa

has a similar structure to PAA and 2,4-D and may function as an auxin analog. Here, we characterized the physiological and molecular activities of Dcaa.

## Results

**Discovery of the Dcaa auxin analog**. As stated in the introduction, existing auxin-like PGRs still seem to have drawbacks and limitations[44,45]. In order to find a more efficient, promising, and safer auxin analog, we designed new compounds based on the chemical structures of existing auxins, such as indole ring (I), naphthalene ring (II), and benzene ring (III) (Fig. 1a). More than 2000 compounds were designed by the "me too" and "active substructure splicing" methods. Based on factors such as the availability of raw materials, difficulty in synthesis, compound stability and toxicity, and cost control, 82 compounds were screened out for further physiological activity tests (Fig. 1b). Through pot experiments and field trials, we ultimately determined that Dcaa (III-7) is a potential and efficient PGR that can promote plant growth (Fig. 1 and Supplementary Data 1).

**Dcaa promotes the elongation of oat coleoptile segments**. To analyze the auxin activity of Dcaa, we performed the classical oat coleoptile segments elongation assay and used NAA, IBA, 2-naphthoxyacetic acid, 1-naphthylacetamide and vitamin B1 (VB1) as controls (Fig. 2a). When treated with 1, 10, and 100 μM Dcaa, the elongation of oat coleoptile segments increased by approximately 27%, 33%, and 83%, respectively, compared with the untreated ones (Fig. 2b, c). By contrast, 1, 10, and 100 μM IBA promoted the elongation by approximately 89%, 165%, and 156%, respectively (Figs. 2b, c), and 10 μM NAA by approximately 202%. Therefore, at the same concentration, IBA and NAA stimulated more elongation than Dcaa. The effect of 1 μM IBA was comparable to that of 100 μM Dcaa. Conversely, VB1, 2-naphthoxyacetic acid, and 1-naphthylacetamide inhibited the elongation (Fig. 2b, c). These results indicate that, like IBA and NAA, Dcaa can promote the elongation of oat coleoptile segments, but its activity is lower than IBA and NAA.

**Dcaa stimulates root development and growth**. An important feature of auxin is to stimulate root development and growth[39,41]. We therefore studied the effect of Dcaa on adventitious root generation by using the hypocotyl of mung bean seedlings and, NAA was used as a control. The results show that Dcaa treatment increases the length of the rooting zone and the number of adventitious roots generated at the base of the hypocotyl of mung bean seedlings in a dose-dependent manner, while the length of the adventitious roots increases at low concentrations of Dcaa but decreases at high concentrations (Fig. 3a, b). Overall, the effect of 120 ppm Dcaa is comparable to that of 25 ppm NAA (Fig. 3a, b).

We next analyzed the effect of Dcaa on the root growth of various crops, including cucumber, cabbage, tomato, and maize. 1.5, 3, 6, and 12 ppm Dcaa were used to irrigate cucumber seedlings at the stage of two euphylla, and a mixture of potassium indole butyrate and sodium naphthylacetate was used as a control (ck). Seven days post-treatment, the height of cucumber seedlings treated with Dcaa and ck was significantly lower than that of the untreated group, whereas the fresh and dry weights of the root systems increased with the increase in the concentration of Dcaa (Fig. 4a–c). The effect of 12 ppm Dcaa on root fresh/dry weight was comparable to that of the ck (Fig. 4c). Notably, the total length and surface area of the root system of the cucumber seedlings reached the highest at the concentration of 3 ppm Dcaa (approximately 252.83 cm and 36.64 cm$^2$, respectively), which was significantly longer/larger than that of the untreated and ck group (untreated: 194.01 cm and 27.95 cm$^2$, respectively; ck:

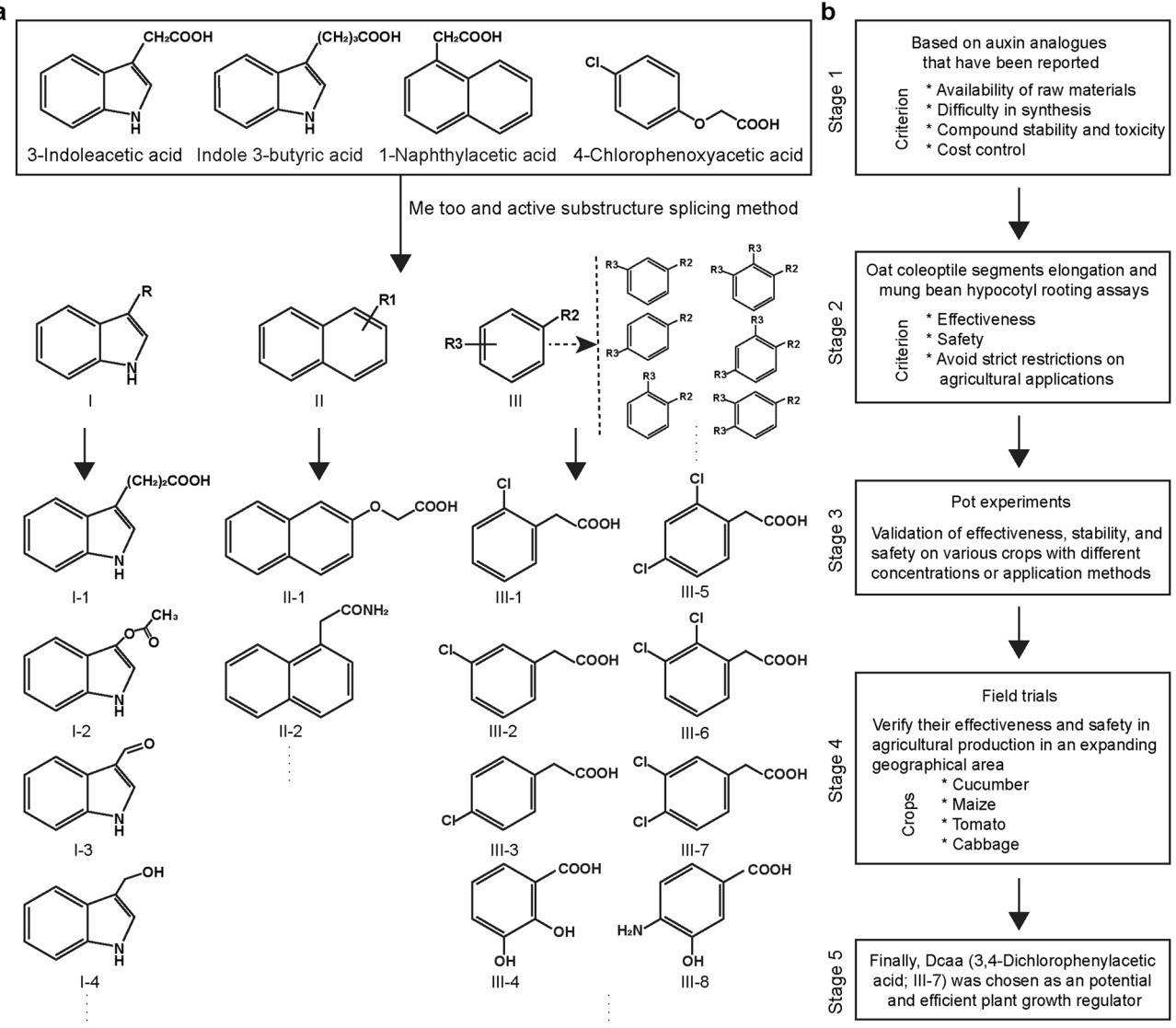

**Fig. 1 Design and screening of auxin analogs. a** Chemical structure of different auxins and analogs. Indole ring (I): 3-indolepropionic acid (I-1), 3-acetoxyindole (I-2), indole-3-carboxaldehyde (I-3), indole-3-methanol (I-4); naphthalene ring (II): 2-naphthoxyacetic acid (II-1), 1-naphthylacetamide (II-2); benzene ring (III): 2-chlorophenylacetic acid (III-1), 3-chlorophenylacetic acid (III-2), 4-chlorophenylacetic acid (III-3), 2,3-dihydroxybenzoic acid (III-4), 2,4-dichlorophenylacetic acid (III-5), 2,3-dichlorophenylacetic acid (III-6), 3,4-dichlorophenylacetic acid (III-7), 4-amino-3-hydroxybenzoic acid (III-8). R represents different substituents on the indole ring. R1 represents the substituents at different positions on the naphthalene ring. R2 represents carboxyl groups containing 0-3 methylene groups on the benzene ring, while R3 represents substituents at different positions on the benzene ring. **b** Flow chart for screening auxin-like PGRs with potential efficacy, including five stages.

194.94 cm and 30.37 cm$^2$, respectively) (Fig. 4d, e). Similarly, the root diameter and total root volume also increased with the increase in the concentration of Dcaa, and reached the highest at the concentration of 12 ppm (Fig. 4f, g). By contrast, there was no significant difference in the number of root branches between the treated and untreated ones (Fig. 4h). These results indicate that Dcaa promotes root elongation and thickening but not branching in cucumber. In brief, the growth of the cucumber root system treated with 3 ppm Dcaa was more vigorous in all root indexes than that under the ck treatment. These results show that Dcaa has a beneficial effect on promoting cucumber root growth and, the effective concentration is 3 ppm.

We also analyzed the effect of Dcaa on the growth of cabbage roots. Similarly, the height of cabbage seedlings treated with a low concentration of Dcaa (1.5 ppm) was significantly lower than that of the untreated group (Supplementary Fig. 1a). However, the fresh and dry weights of the cabbage root system increased with

the increase in the concentration of Dcaa and, were markedly higher than that of the ck treatment (Supplementary Fig. 1b, c). The total length, branch number, and total surface area of the cabbage root system increased with the increase in the concentrations of Dcaa, but there were no statistical differences (Supplementary Fig. 1d–f). By contrast, the diameter and total volume of the cabbage roots under the 12 ppm Dcaa treatment were significantly larger than that of the untreated group (Supplementary Fig. 1g, h), indicating that it could promote the radial growth of the cabbage root. Notably, the cabbage roots under Dcaa treatments performed better than those under the ck treatment. These results indicate that Dcaa has a promoting effect on cabbage root growth.

To study the effect of Dcaa on the roots of monocotyledonous crops, we performed root growth experiments on maize by irrigation with different concentrations of Dcaa. Compared to the untreated, Dcaa treatment significantly increased the fresh/dry

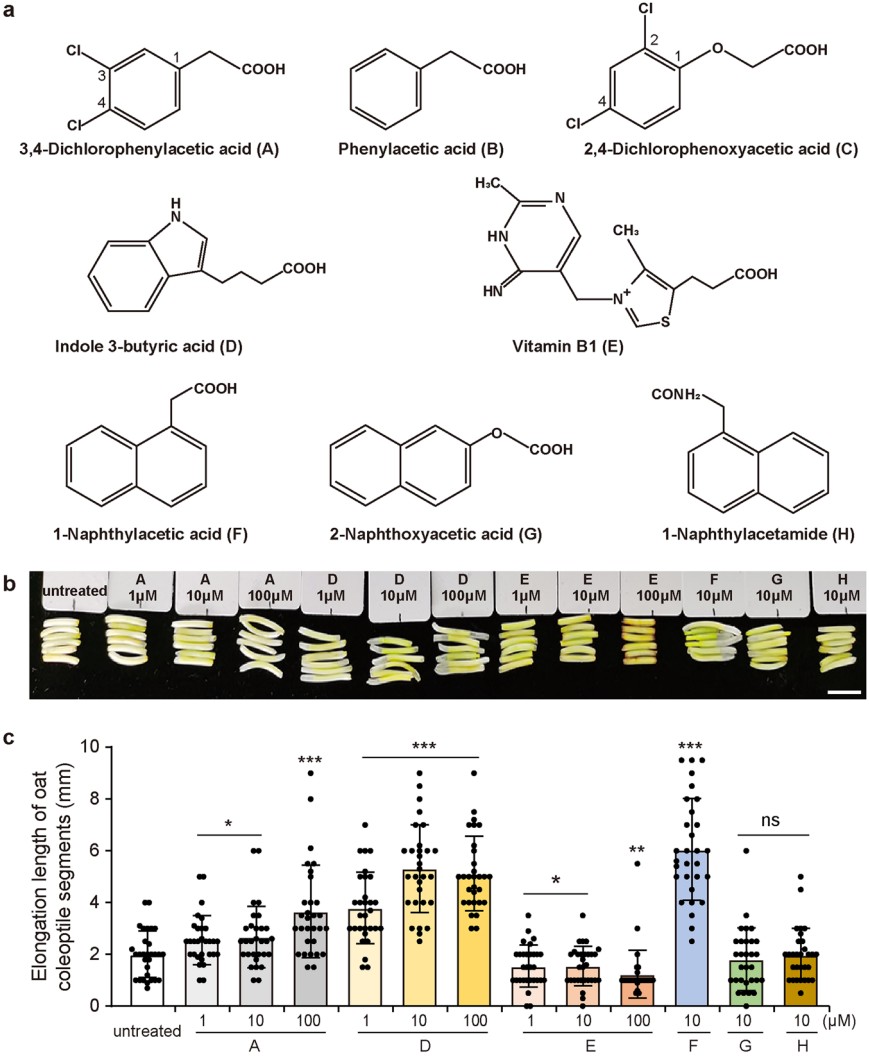

**Fig. 2 3,4-dichlorophenylacetic acid (Dcaa) promotes the elongation of oat coleoptile segments. a** Chemical structures of different PGRs. **b**, **c** The representative images (**b**) and elongation length (**c**) of oat coleoptile segments 40 h after treatment with various concentrations of PGRs. Note that A, D, E, F, G, and H represent the PGRs displayed in **a**, respectively. Approximately employing 30 oat coleoptile segments per treatment. *$P < 0.05$, **$P < 0.01$, ***$P < 0.001$, *ns* not significant (Student's *t*-test, two-tailed, two-sample equal variance). Bar = 1 cm (**b**).

weight, total length, and total surface area of the maize root system (Supplementary Fig. 2a–e). By contrast, the branch number, diameter, and total volume of the maize root system under the Dcaa treatment were not significantly different from that of the untreated (Supplementary Fig. 2f–h). Although the ck treatment significantly increased the total length, surface area, and volume of the maize root system, its effect on root fresh/dry weight was lower than that of the Dcaa treatment, probably because it stimulated fewer branches than Dcaa. These results indicate that Dcaa can effectively promote root growth of monocot crops.

Irrigation roots with different concentrations of Dcaa can effectively promote the growth of crop roots. Subsequently, we analyzed the effects of spraying leaves with Dcaa on the growth of tomato roots. Similarly, we used the mixture of potassium indole butyrate and sodium naphthalene acetate compounds as a control. Although the total length of tomato roots was not statistically different from that of the untreated group (61.2 cm vs 44.7 cm), the total projected area, surface area, and volume of the tomato root system were significantly different from those of the untreated group under the treatment of 6 ppm Dcaa (Supplementary Fig. 3). Unlike the root irrigation of cucumber and

cabbage plants, leaf spraying of Dcaa did not increase the root diameter of tomato plants (Supplementary Fig. 3e). Despite this, we still can conclude that 6 ppm of Dcaa is a reasonable concentration to promote tomato root growth.

Therefore, whether it is applied by root irrigation or foliar spraying, Dcaa can promote the growth of crop roots.

**Dcaa promotes nitrogen fertilizer use efficiency in maize**. It has been shown that auxin plays an important role in promoting plant nitrogen use efficiency[46–48]. To analyze whether Dcaa can promote nitrogen use efficiency in plants, we conducted an assay on maize. We treated maize seedlings with 3, 6, and 12 ppm Dcaa, respectively, either by spraying leaves or by irrigating roots (Supplementary Fig. 4a). Nine days post-treatment, the nitrogen content in the aboveground tissues of maize seedlings treated with various concentrations of Dcaa was significantly higher than that of the untreated group (Supplementary Fig. 4b). The nitrogen use efficiency under 3 ppm Dcaa treatment by spraying leaves reached 34.7% and decreased with the increase of concentration (Supplementary Fig. 4c). By contrast, the treatment of root irrigation exhibited the highest promotion rate at 6 ppm Dcaa

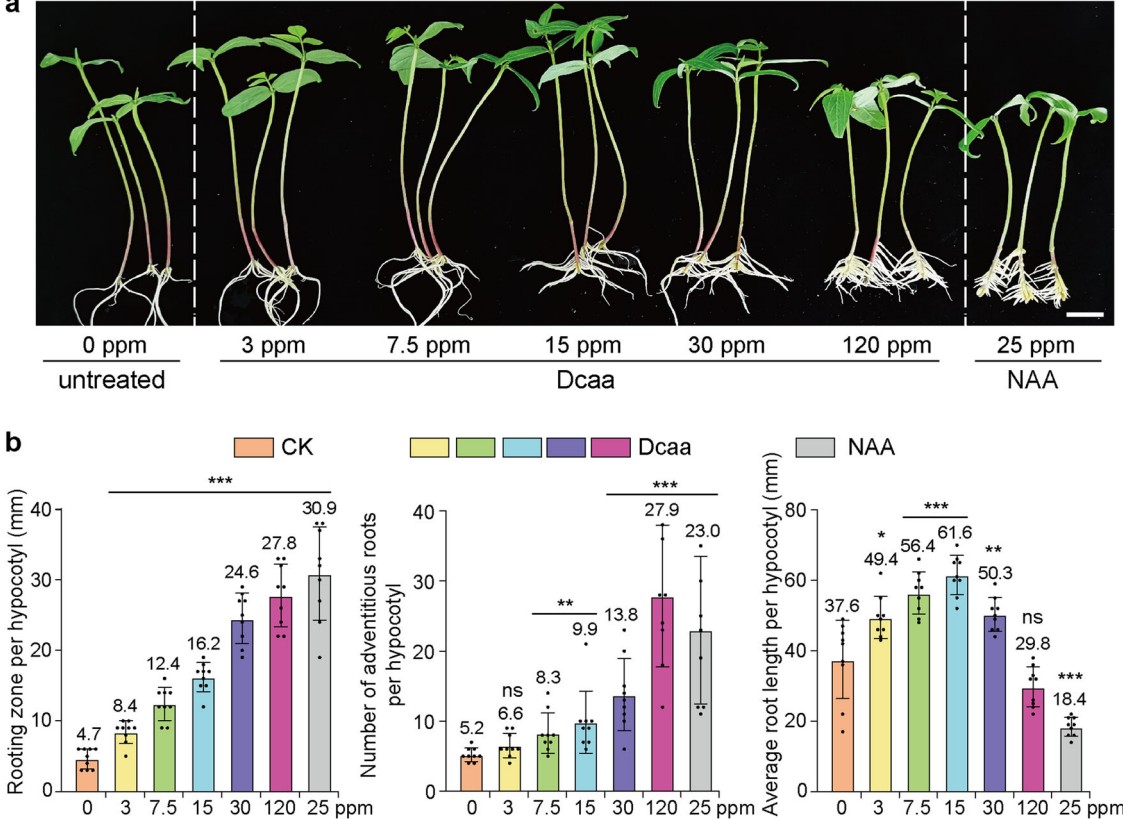

**Fig. 3 Dcaa promotes the production of adventitious roots from the hypocotyl of mung bean seedlings. a** Representative images showing adventitious roots generated at the base of the hypocotyls of mung bean seedlings 7 days after being treated with various concentrations of Dcaa or NAA for 24 h. Bar = 2 cm. **b** Average length of the rooting zone and number and length of adventitious roots produced per hypocotyl. Error bars represent the SD of the mean of 9 hypocotyls. *$P < 0.05$, **$P < 0.01$, ***$P < 0.001$, *ns* not significant (Student's *t*-test, two-tailed, two-sample equal variance).

(Supplementary Fig. 4c). These results indicate that Dcaa can promote nitrogen use efficiency by maize.

**Dcaa regulates the expression of auxin-responsive reporters and genes**. To determine whether Dcaa acts through the auxin signaling pathway, we first analyzed its effect on the expression of the auxin responsive-reporter *DR5:GUS* and *DR5rev:GFP* and used NAA as a control. The *DR5:GUS* seedlings displayed GUS signals in the quiescent center (QC) and columella cells of the radicle and lateral root tip and in the root vascular tissues (Fig. 5a). After 10 μM NAA treatment, the GUS signals were markedly enhanced, and additional signals appeared at the stele of the root tip (Fig. 5b). Similarly, 250 and 500 μM Dcaa treatments also enhanced GUS signals in the quiescent center, columella cells, and stele of the root tip (Fig. 5c, d). We further quantified the activity of the GUS enzyme. The results showed that after Dcaa and NAA treatment, the GUS enzyme activity was significantly higher than that of the untreated group (Fig. 5e; $P < 0.01$). The activity of the GUS enzyme was also significantly increased with the increase in the concentration of Dcaa (Fig. 5e; $P < 0.01$). Similarly, we treated *DR5rev:GFP* Arabidopsis transgenic seedlings with 10 μM NAA and 2,4-D and 500 μM Dcaa, respectively. The GFP fluorescent signal intensity was significantly enhanced in the quiescent center, columella, and stele cells compared with the untreated group (Supplementary Fig. 5).

We next performed reverse transcription-quantitative polymerase chain reaction (RT-qPCR) analyses to determine whether Dcaa could regulate the expression of auxin-responsive genes. We selected six genes: *ARF7*, *ARF19*, *IAA19*, *LBD16*, *SAUR22*, and

*SAUR24*, which rapidly respond to auxin and participate in root growth and development. The results showed that under treatment with low concentrations of NAA (1 μM) and Dcaa (100 and 250 μM), the expression of *ARF7* decreased over time (Fig. 6a), while the expression of *LBD16* and *SAUR24* increased over time (Fig. 6d, f). The expression of *SAUR22* first increased and then decreased (Fig. 6e), while *ARF19* showed no significant changes (Fig. 6b). Under 1 μM NAA treatment, the expression of *IAA19* increased over time, while under 100 and 250 μM Dcaa treatment, its expression first increased and then decreased (Fig. 6c). Under treatment with high concentrations of NAA (10 μM) and Dcaa (500 μM and 1 mM), the expression levels of *ARF7*, *ARF19*, *IAA19*, and *SAUR22* showed a trend of first increasing and then decreasing (Fig. 6a–c, e), while the expression of *LBD16* and *SAUR24* genes increased over time (Fig. 6d, f). Overall, the expression of these genes showed similar responses to Dcaa and NAA treatments.

Taken together, these results indicate that Dcaa can regulate the expression of auxin-responsive reporters and genes.

**Dcaa can be perceived by auxin receptors**. To verify whether Dcaa could be perceived by the auxin receptors, we treated auxin receptor mutants *tir1-1* and *tir1 afb1,2,3* with various concentrations of Dcaa and, 2,4-D was used as a control. The growth of the wild-type roots was strongly inhibited under 2,4-D and Dcaa treatment (Fig. 7). Although the growth of *tir1-1* and *tir1-1 afb1,2,3* roots was also inhibited by 2,4-D and Dcaa treatments due to the redundancy of the auxin receptors in Arabidopsis (there are 6 auxin receptors including TIR1 and AFB1-5 in

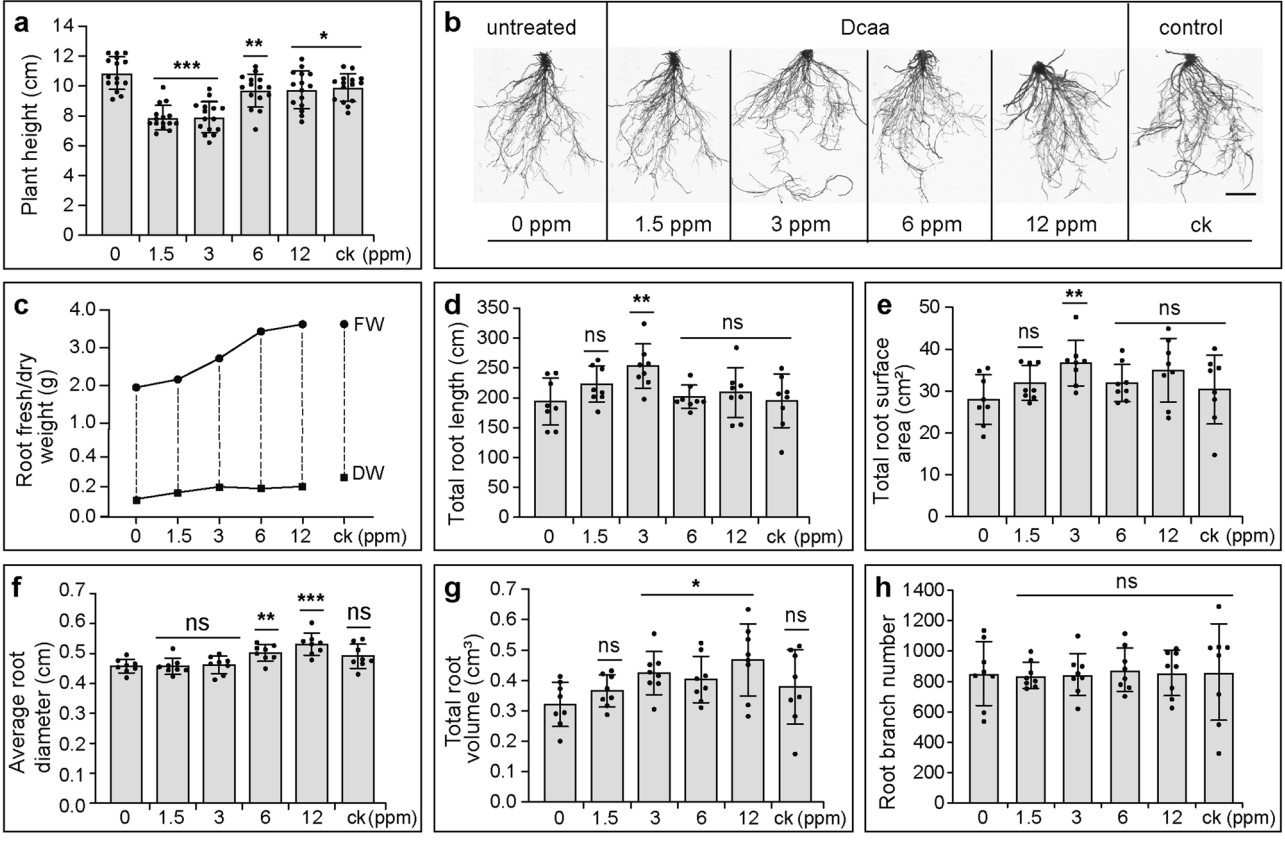

**Fig. 4 Dcaa promotes the growth of cucumber roots. a**, **b** The average height (**a**) and representative images of the root system (**b**) of cucumber plants 7 days after being treated with various concentrations of Dcaa and a PGR composed of 1 ppm potassium indole butyrate and 1 ppm sodium naphthalene acetate (ck). **c**–**h** Root fresh/dry weight (**c**), total root length (**d**), total root surface area (**e**), average root diameter (**f**), total root volume (**g**), and root branch number (**h**). Error bars represent the SD of the mean of 8–15 cucumber plants. *$P < 0.05$, **$P < 0.01$, ***$P < 0.001$, *ns* not significant (Student's *t*-test, two-tailed, two-sample equal variance). Bar = 2 cm (**b**).

Arabidopsis), the inhibition degree was significantly weaker than that of the wild-type seedlings (Fig. 7), suggesting that Dcaa acts through these auxin receptors.

Crystal structure analysis showed that the TIR1 auxin receptor protein could potentially bind various auxin compounds[49]. Although the chemical structure of the synthetic auxins, such as NAA and 2,4-D, is different from that of the plant endogenous auxin IAA, they can bind to the TIR1-LRR (leucine-rich-repeat) domain[49]. To further analyze the perception of Dcaa by the TIR1 and AFB1-5 auxin receptors, we carried out molecular docking studies. The binding energies between Dcaa and TIR1 or AFB1-5 are calculated to be -6.037, -5.503, -5.412, -5.149, -5.620, and -5.610, respectively, suggesting that Dcaa is perceived preferentially by the TIR1 protein. Similar to IAA, the carboxyl group of the Dcaa molecule can form favorable hydrogen bonding interactions with the two highly selective polar residues Arg 403 and Ser 438 of the TIR1 substrate binding pocket (Fig. 8). Additionally, Dcaa can also form a hydrogen bond with the Arg 436 of TIR1 (Fig. 8). These interactions presumably promote the anchoring of Dcaa to the TIR1 ligand binding pocket. Notably, the amino acid residues interacting with Dcaa of the AFB1-5 proteins simulated by homologous modeling were different from that of the TIR1 protein (Fig. 8). The binding ability of AFB4 and AFB5 to Dcaa was higher than the other AFB proteins.

Together, these results suggest that Dcaa can be perceived by the auxin receptors.

**Roots of *pin2-T* but not *aux1-T* mutants are hypersensitive to Dcaa treatment.** A distinct feature of auxin is that it is

transported directionally in plants[6,9]. To determine whether Dcaa could be polarly transported in plants, we carried out an auxin application assay. Agar gel strips containing Dcaa, IAA, 2,4-D, and NAA were placed at the root tips of 5-day-old *DR5:GUS* and *DR5rev:GFP* Arabidopsis seedlings, respectively (Fig. 9a and Supplementary Fig. 6a). The transport of these substances can be indirectly indicated by the enhanced GUS and GFP signals intensity in the root elongation and maturation zone. The results showed that compared to the mock treatment, the *DR5:GUS* signals were markedly increased in the root tip and root hair zone after IAA, 2,4-D, NAA, and Dcaa treatment (Fig. 9a). Consistently, after IAA and Dcaa treatment, the *DR5rev:GFP* signals in the root stele were markedly enhanced (Supplementary Fig. 6b). Further quantitative analysis of GFP fluorescence intensity showed that the GFP signal intensity was significantly increased after IAA and Dcaa treatment (Supplementary Fig. 6c). These results imply that similar to IAA, Dcaa may be transported basipetally in the root. However, we cannot exclude that the enhanced expression of *DR5:GUS/DR5rev:GFP* was due to the sustained presence of the induced auxin signaling.

It is well known that the movement of IAA across the plasma membrane requires both influx and efflux carriers, whereas 2,4-D requires influx carriers but is hardly exported by efflux carriers and, NAA requires efflux carriers but not influx carriers[50,51]. Thus, auxin influx carrier mutants (such as *aux1*) are resistant to 2,4-D but not NAA, whereas auxin efflux carrier mutants (such as *pin2*) are hypersensitive to NAA but not 2,4-D[51]. To determine the effect of Dcaa on auxin transport mutants, we first analyzed the expression of the *DR5:GUS* reporter in *aux1-T* and *pin2-T*

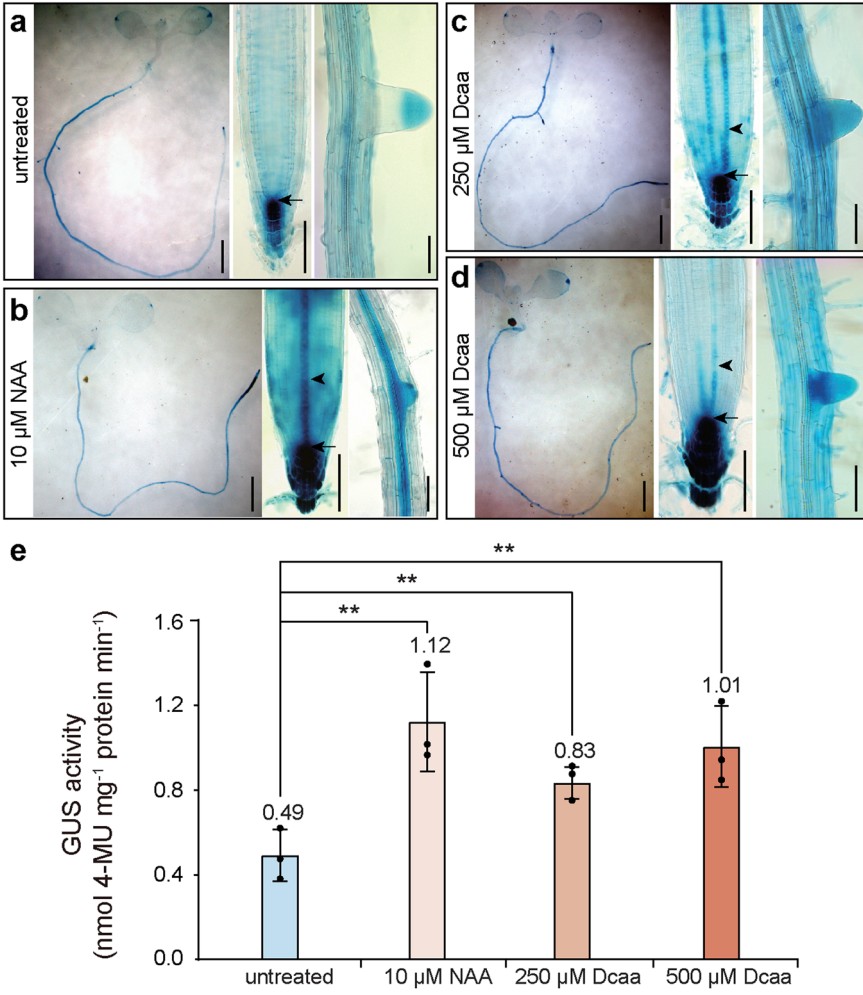

**Fig. 5 Dcaa enhances the expression of the *DR5:GUS* auxin-responsive reporter. a–d** Histochemical staining for GUS activity in 7-day-old Arabidopsis seedlings of untreated (**a**) or treated for 2 h with 10 μM NAA (**b**) and 250 (**c**) and 500 μM (**d**) Dcaa. In each panel from left to right are seedling, root tip, and lateral root. Arrows indicate GUS signals at the quiescent center (QC) and arrowheads indicate GUS signals at the stele of the root tip. Bars = 1 mm (seedlings) and 100 μm (root tips and lateral roots). **e** Quantitative GUS activity assay of 7-day-old seedlings. Error bars represent the SD of the mean of three biological experiments (30 seedlings per treatment). **$P < 0.01$ (Student's *t*-test, one-tailed, two-sample equal variance).

mutant roots after the application of Dcaa at the root tip[52]. In *pin2-T DR5:GUS* roots, except for 2,4-D, application of IAA, NAA, and Dcaa at the root tip did not result in enhanced GUS signals in the root hair zone (Fig. 9b), suggesting that like IAA and NAA, intercellular transport of Dcaa may require efflux carriers. By contrast, in *aux1-T DR5:GUS* roots, NAA and Dcaa but not IAA and 2,4-D caused an increase in *DR5:GUS* signal in the root hair zone (Fig. 9c), suggesting that like NAA, intercellular transport of Dcaa does not require an influx carrier.

We next tested the sensitivity of the *aux1-T* and *pin2-T* mutants to Dcaa and, NAA and 2,4-D were used as controls. The results showed that the *aux1-T* mutant was remarkably resistant to 2,4-D treatment but displayed similar responses as the wild type to NAA and Dcaa treatments, whereas the *pin2-T* mutant was hypersensitive to NAA and Dcaa treatments but exhibited similar responses as the wild type to 2,4-D (Fig. 9d, e). These results again imply that Dcaa may be a substrate of PIN2 but not AUX1.

Together, these results imply that Dcaa can enter the cell via diffusion but may require carrier-mediated efflux.

**Dcaa inhibits endocytosis of PIN efflux carrier proteins**. It has been shown that the internalization of PIN proteins mediated by

the vesicle trafficking inhibitor brefeldin A (BFA) decreased significantly under the co-treatment of auxin and BFA[12,13]. To further analyze whether Dcaa could inhibit the internalization of PIN proteins, we employed NAA, 2,4-D, and Dcaa with BFA to co-treat *PIN2-GFP* seedlings. After BFA treatment, PIN2-GFP formed approximately 1 to 3 BFA bodies inside the cell, whereas co-treatment with Dcaa, NAA, and 2,4-D, significantly reduced the number of BFA bodies inside the cells (Fig. 10), indicating that, like other auxin analogs, Dcaa can inhibit the endocytosis of PIN2 protein.

## Discussion

In this study, we screened for new auxin-like compounds that can be used in broad species of crops. We identified that Dcaa has auxin-like activity and can promote the development of the root system of various crops including cucumber, cabbage, tomato, and maize. The results of coleoptile segment elongation and adventitious root generation experiments indicate that the auxin activity of Dcaa is lower than that of IBA and NAA. However, when applied to intact crop plants, particularly cucumber and cabbage, Dcaa induced higher growth-promoting effects on roots than the mixture of potassium indole butyrate and sodium naphthalene acetate. This discrepancy may be due to that

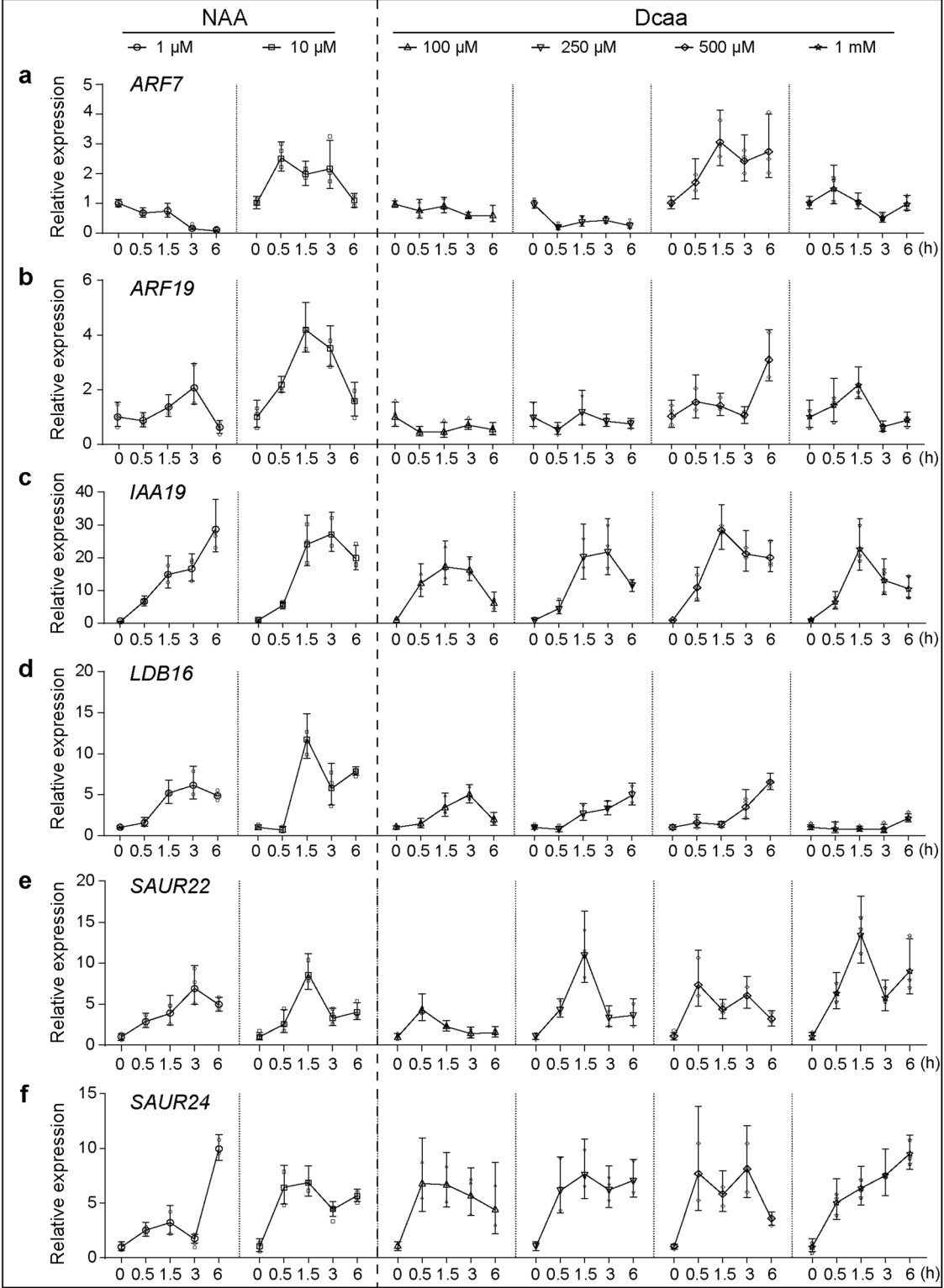

**Fig. 6 Dcaa induces the transcription of auxin-responsive genes. a–f** Relative transcription levels of the *ARF7*, *ARF19*, *IAA19*, *LBD16*, *SAUR22*, and *SAUR24* genes in 7-day-old Arabidopsis seedlings after 0, 0.5, 1.5, 3, and 6 h treatment with various concentrations of Dcaa and NAA. The *TIP41* gene was used as an internal control. The experiment was performed by three biological repeats and shown are the results of one representative repeat. Error bars represent the SD of three technical replicates.

coleoptile segments and mung bean hypocotyls were immersed in the auxin solutions, which may provide a more direct uptake of the auxin[50]. The physiological activity of auxin analogs in intact plants is determined by their chemical and metabolic stability, their ability to be transported directionally in tissues, and their

activity toward auxin signaling components[24,35]. On the other hand, since the root is the most sensitive plant organ to auxin, exogenous application of auxins usually produces an inhibitory effect on root elongation[35,53]. Therefore, auxins with higher activity may be more likely to have inhibitory rather than

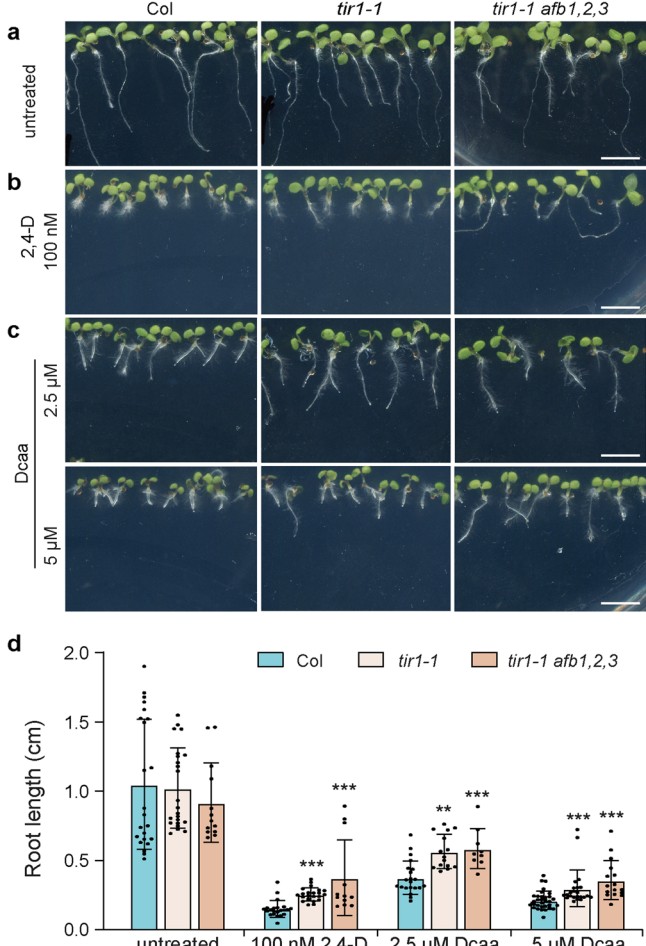

**Fig. 7 *tir1-1* and *tir1-1 afb1,2,3* seedlings are less sensitive to Dcaa and 2,4-D. a–c** Auxin receptor mutants *tir1-1* and *tir1-1 afb1,2,3* are less sensitive to Dcaa and 2,4-D treatments compared to Col. **d** The root length of 7-day-old Col, *tir1-1*, and *tir1-1 afb1,2,3* seedlings. Error bars represent the SD of the mean of 12-33 seedlings. **\*\***$P < 0.01$, **\*\*\***$P < 0.001$ (Student's *t*-test, two-tailed, two-sample equal variance). Bars = 2 mm.

Arg436 residue, suggesting that the dichlorophenyl ring of Dcaa may fit into the hydrophobic cavity of the TIR1 pocket with a slightly different orientation than that of 2,4-D. This is presumably due to the position of the two chlorines and the length of the side chain of Dcaa and 2,4-D being different.

A unique feature of auxin is its polar transport in plants. Long-distance transport of auxin, i.e., from the shoot apical bud towards the root tip is carried out through the phloem. Short-distance cell-to-cell transport of auxin is mediated by the plasma membrane-localized AUX1/LAX auxin influx carrier proteins and PIN efflux carrier proteins[6,8,9]. PIN proteins are polarly localized to one side of the cell and mediate the direct flow of auxin in plant tissues[6,10,13]. The major plant endogenous auxin IAA enters the cell by passive diffusion and influx carrier-mediated active transport, but it can only be transported out of the cell by the auxin efflux carrier proteins[50]. Thus IAA is transported actively and directionally in plants and generates concentration gradients in plant tissues and organs in response to developmental and environmental signals[30]. By contrast, the other plant native auxin PAA is not polarly transported in plants[30]. The synthetic auxin analog NAA enters the cell mainly by passive diffusion but is exported out of the cell by efflux carrier-mediated active transport, whereas 2,4-D depends on influx carriers to enter the cell but can hardly be transported out of the cell by either diffusion or efflux carrier-mediated active transport[50,51]. Our data suggest that Dcaa may be polarly transported in plants and, its transport property is similar to that of NAA, i.e., Dcaa enters the cell mainly by passive diffusion but may require carrier-mediated efflux. However, our data also indicates that *aux1-T* roots are more resistant to Dcaa and NAA treatments compared to that of the wild type (Fig. 9d, e), suggesting that Dcaa and NAA may also enter cells through the AUX1 influx carrier besides passive diffusion. Previously, Imhoff et al. analyzed the effects of several chemicals on auxin influx in suspension-cultured tobacco cells by the accumulation of labeled [$^{14}$C]2,4-D[54]. By measuring the accumulation of [$^{14}$C]2,4-D in tobacco cells in the presence of various concentrations of phenylacetic or phenyloxyacetic acid derivatives, it was found that the substitution of atoms on the aromatic ring of most phenylacetic and phenyloxyacetic acids significantly reduced the influx of [$^{14}$C]2,4-D, especially Dcaa and 3,4-dichlorophenoxyacetic acid[54]. Similarly, 3-chloro-4-hydroxyphenylacetate acid also showed strong inhibition of auxin influx activity[54–56]. As discussed by the authors, phenylacetic acid derivatives are able to inhibit auxin influx because they may interact with the influx carrier and compete with 2,4-D for auxin-binding sites on the influx carrier, suggesting that these derivatives that inhibit auxin influx are better matched to the influx carriers than 2,4-D[54]. Recently, the structure of PIN1, 3, and 8 proteins has been solved. The intracellular substrate pocket of PIN proteins accommodates the auxin molecule by hydrophobic stacking and hydrogen bonding and, selects for auxin molecules on the basis of shape complementarity[36,57,58]. The auxin-efflux transporter inhibitor N-1-naphthylphthalamic acid (NPA) competes with IAA for the same pocket but has a higher affinity and locks the PIN proteins in the substrate binding conformation[36,57,58]. Compared with IAA, the larger size of IBA and the small ring of PAA made them less fit in the cavity of the PIN substrate pocket[58]. Although Dcaa, like PAA, has a small ring structure, the two large chlorines may enable it to mimic the double rings of IAA and NAA.

stimulatory effects on root growth. For example, in cabbage, 1.5 to 12 ppm Dcaa induced growth-promoting effects on the root, whereas the mixture of 1 ppm potassium indole butyrate and 1 ppm sodium naphthalene acetate inhibited root growth.

Our results show that Dcaa acts through the auxin signaling machinery. In Arabidopsis, auxin is perceived by a family of F-box proteins including TIR1 and AFB1-5[14,15]. The crystal structure analysis has revealed that the auxin-binding site of TIR1 is mainly driven by two kinds of interactions. One is composed of two highly selective polar residues (Arg 403 and Ser 438), which interact with the carboxyl group of auxin molecules through the salt bridge and hydrogen bond. The other is a hydrophobic cavity with a fixed shape that binds the ring of auxin molecules through hydrophobic interactions and van der Waals contacts, which is relatively less selective[49]. Thus, TIR1 can bind auxin molecules with different rings such as the indole ring of IAA, the naphthalene ring of NAA, and the dichlorophenyl ring of 2,4-D, which all fit well into the hydrophobic cavity of TIR1 with an almost identical orientation[49]. Like 2,4-D, Dcaa also has a dichlorophenyl ring, suggesting that it may bind to the TIR1 pocket in a similar manner. Indeed, our molecular docking data show that similar to 2,4-D, the carboxyl group of Dcaa can form hydrogen bonds with the Arg 403 and Ser 438 residues of the TIR1. However, Dcaa can additionally form hydrogen bonds with the

## Methods

**Plant materials and growth conditions.** The *Arabidopsis thaliana* mutants and reporters used have been described previously: *aux1-T*[59], *pin2-T*[11], *DR5:GUS*[4], *DR5rev:GFP*[5], *ProPIN2:PIN2-*

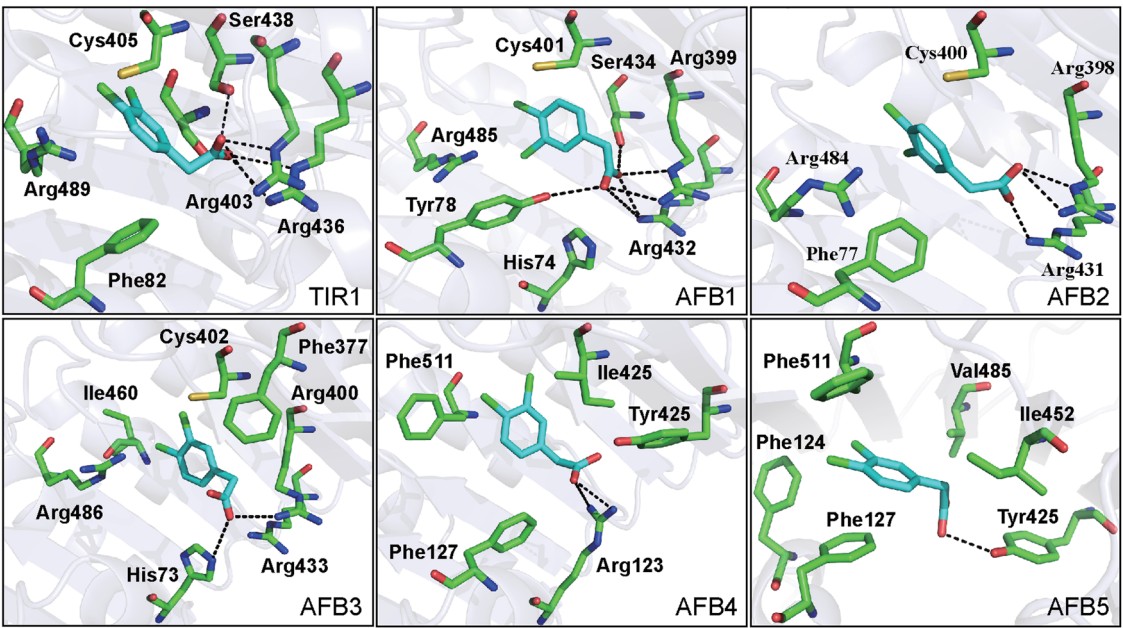

**Fig. 8 Binding of Dcaa by the TIR1 and AFB1-5 auxin receptors.** Molecular modeling of the interaction between Dcaa (cyan) and the TIR1 and AFB1-5 auxin receptors, with key residues of the main binding pocket shown with colored heteroatoms. Interatomic distances are shown as dashed black lines.

GFP[60], *tir1-1* and *tir1-1 afb1,2,3*[14]. The Col-0 (Col) ecotype was used as a wild-type control. Seeds were surface sterilized for 5 min in 70% (v/v) ethanol and 10 min in 1% (v/v) Clorox bleach, and then washed five times with sterile water. After stratification for 2–3 days at 4 °C, seeds were sown on Murashige and Skoog (MS) medium and cultured under a 16 h light/8 h dark cycle at 22 °C.

Oat (*Avena sativa L.*) seeds were sown in soil and cultured under a 12 h light/12 h dark cycle at 22 °C in an incubator. Mung bean cv. Jizaolvzhenzhu 2 seeds after being soaked in water for 24 h were grown in soil with a 12 h light/12 h dark cycle at 25 °C in an incubator. Seeds of cabbage cv. Xiaguan F1, cucumber cv. Jinpei 98F1, tomato cv. Kaideyali 1832, and maize cv. Yudan 9953 were grown in soil with a 12 h light/12 h dark cycle at 25 °C in a greenhouse.

**Chemicals**. NAA, IAA, 2,4-D, IBA, naphthalene acetamide, naphthoxyacetic acid, vitamin B1, potassium indole butyrate, and sodium naphthalene acetate were purchased from Aladdin (Shanghai, China).

**Oat coleoptile segments elongation assay**. Oat coleoptiles germinated to 2–3 cm were cut from the top to 1 cm as experimental materials. Then 30 oat coleoptile segments were immersed in different concentrations of Dcaa (1, 10, and 100 μM), NAA (10 μM), IBA (10 μM), naphthalene acetamide (10 μM), naphthoxyacetic acid (10 μM), and vitamin B1 (10 μM) solutions in dark condition, respectively. The length of oat coleoptile segments was measured after 40 h treatment by the ImageJ software (http://rsb.info.nih.gov/ij)[61].

**Mung bean hypocotyl adventitious roots generation assay**. Mung bean seedlings were cultured to about 15 cm, and then the roots were cut off at an inclined plane and soaked in 25 ppm NAA or 3, 7.5, 15, 30, and 120 ppm Dcaa solution, respectively. After 24 h treatment, the shoots were put into water and continued to be cultured for one week. Then the length of the rooting zone and the number and length of adventitious roots were scored.

**Crops root growth assay**. Cucumber and cabbage seedlings at the stage of two euphylla were treated with 5 mL Dcaa at concentrations of 1.5, 3, 6, and 12 ppm, respectively, in the manner of root irrigation. At the same time, the mixture of 1 ppm potassium indole butyrate and 1 ppm sodium naphthylacetate was used as a control. The root systems of cucumber and cabbage seedlings were analyzed after 7 days.

Tomato seedlings cultured to 4–5 cm were treated with different concentrations of Dcaa (3, 6, and 12 ppm) by foliar spraying. At the same time, a PGR composed of 1:1 of potassium indole butyrate: sodium naphthalene acetate at the concentrations of 10, 20, and 40 ppm, respectively, was used as a control. The roots of tomato seedlings were analyzed after 7 days.

Maize seedlings at the stage of one euphylla were treated with 5 mL of Dcaa at the concentration of 3, 6, and 12 ppm, and the mixture of 1 ppm potassium indole butyrate and 1 ppm sodium naphthylacetate, respectively, in the manner of root irrigation. After 7 days, the roots of maize seedlings were analyzed.

For root analysis, the WinRHIZO root analysis system (Regent Instruments Inc., Québec, Canada) was used. The fresh and dry weight of roots, average root diameter, total root length, volume, projected area, and surface area were scored.

**Maize nitrogen fertilizer use efficiency assay**. Maize seedlings were grown to 10 cm on a substrate supplemented with 8.75 g/L urea and then treated with 3, 6, and 12 ppm Dcaa, respectively, in the manner of spraying leaves and root irrigation with the same dosage. A week later, performed a second treatment in the same manner. After two days, the total nitrogen content in the maize seedling shoots was analyzed.

**DR5:GUS signal intensity and GUS enzyme activity assay**. GUS staining and observation were carried out as previously described[62]. Briefly, 7-day-old *DR5:GUS* Arabidopsis seedlings were treated with 10 μM NAA and 250 or 500 μM Dcaa in liquid MS medium (pH 5.8) for 2 h. These seedlings were then incubated in a GUS staining solution (0.5 mg/mL 5-bromo-4-chloro-3-indolyl-β-D-glucuronic acid, 50 mM sodium phosphate, 0.5 mM ferricyanide, and 0.1% (v/v) Triton X-100, pH 7.0) for 3 h

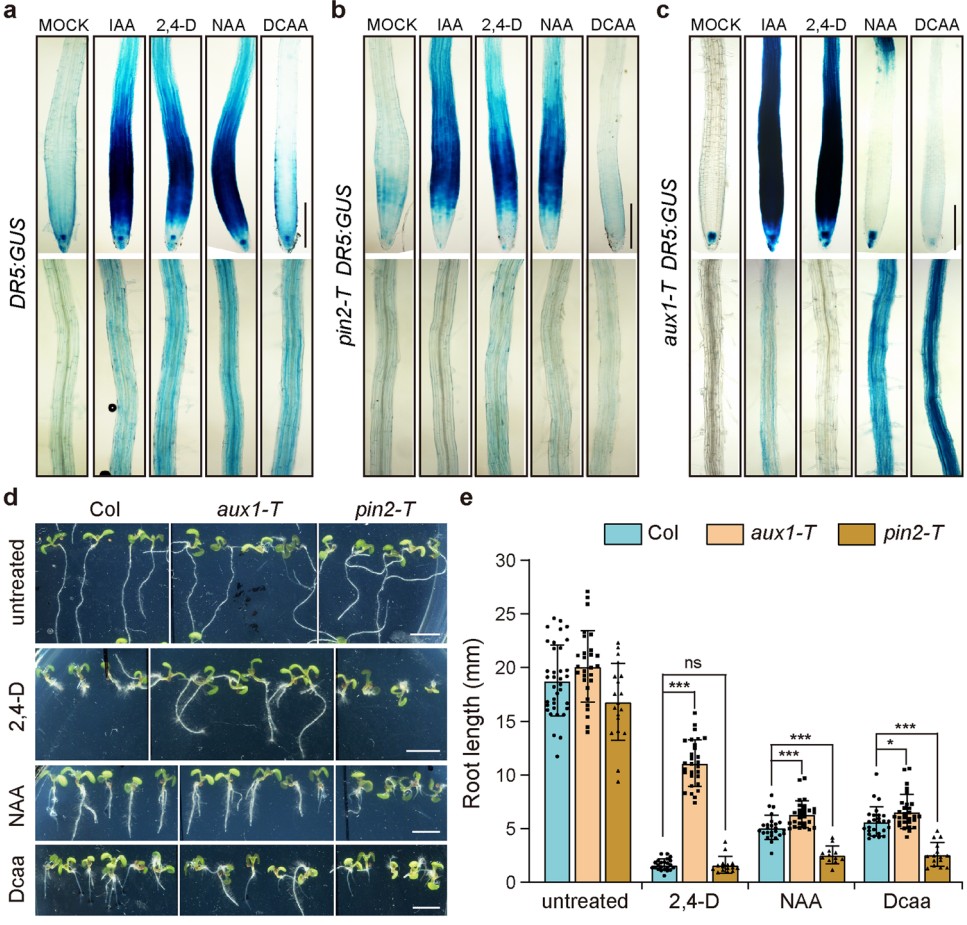

**Fig. 9 Roots of *pin2-T* but not *aux1-T* mutants are hypersensitive to Dcaa treatment. a–c** GUS signals in the primary root of *DR5:GUS*, *pin2-T DR5:GUS*, and *aux1-T DR5:GUS* seedlings 13 h after agar gel strips containing 0 (Mock), 100 μM IAA, 50 μM 2,4-D, 100 μM NAA, or 1.5 mM Dcaa were placed below the root tip (overlapped with the root tip by approximately 0.2 mm). The experiment was performed three times (employed 5-8 roots per treatment per genotype in each experimental replicate) and shown are representative images of one repeat. **d** Auxin transport mutants *aux1-T* and *pin2-T* show different sensitivity compared with Col after 2.5 μM Dcaa, 50 nM 2,4-D, and 200 nM NAA treatment. **e** Quantification of the root length of 7-day-old seedlings of Col, *aux1-T*, and *pin2-T*. Error bars represent the SD of the mean of 12–36 roots. *$P < 0.05$, ***$P < 0.001$, *ns* not significant (Student's *t*-test, two-tailed, two-sample equal variance). Bars = 200 μm (**a–c**) and 5 mm (**d**).

at 37 °C. After incubation, samples were first fixed in ethanol:acetic acid (2:1) for 6 h and then cleared in a clearing solution of chloral hydrate:distilled water:glycerol 8:3:1 (w/v/v). GUS staining signals were observed using a Leica microscope (Leica, Wetzlar, Germany).

GUS enzyme activity assays were performed as previously described[63]. Briefly, 7-day-old *DR5:GUS* Arabidopsis seedlings were treated in the same manner as above, and then 50 mg of the seedlings were collected, quickly frozen in liquid nitrogen, ground into fine powder, and proteins were extracted with 1 mL of extraction buffer (0.1 M $Na_2HPO_4$, 0.1 M $NaH_2PO_4$, 10 mM $Na_2EDTA$, 0.2% (v/v) Triton X-100, 10 mM β-mercaptoethanol). After centrifugation at 12,000 rpm for 10 min, the supernatant was transferred into a new 1.5 mL centrifuge tube. The protein content in the samples was calculated using the BSA protein standard curve. The substrate 4-MUG (4-Methylumbelliferyl-β-D-glucuronidehydrate) was added to the above protein solution. After reacting for 30 min, the fluorescence value of the 4-MU product was measured with a microplate reader (Cytation 5, Bio Tek Instruments, Inc., Vermont, USA). The GUS enzyme activity was calculated through the formula $U = a/(b*t)$ ($U$: enzyme activity, a: 4-MU concentration when reacting for 30 min, b: protein content, t: reaction time).

**DR5rev:GFP reporter assay.** 7-day-old *DR5rev:GFP* Arabidopsis seedlings were treated with 10 μM 2,4-D and NAA and 500 μM Dcaa, respectively, in liquid MS medium (pH 5.8) for 2 h. *DR5rev:GFP* fluorescence signals were collected by a fluorescent microscope (Leica, Wetzlar, Germany) and quantified by the ImageJ software.

**Reverse transcription-quantitative polymerase chain reaction (RT-qPCR).** 7-day-old Arabidopsis wild-type (Col) seedlings were treated with 1 μM or 10 μM NAA, 100 μM, 250 μM, 500 μM, or 1 mM Dcaa in liquid MS medium for 0, 0.5, 1.5, 3, and 6 h, respectively, and then 100 mg of the seedlings were collected, quickly frozen in liquid nitrogen, ground into fine powder, and total RNAs were extracted using TRIzol reagent (TransGen Biotech, Beijing, China). 2 μg total RNAs were reverse transcribed using the EasyScript First-Strand cDNA Synthesis SuperMix kit (TransGen Biotech, Beijing, China). qPCR was performed on a CFX96 real-time PCR detection system (Bio-Rad, Hercules, CA, USA). Each reaction contained 1 μL cDNA, 0.5 μL each of forward and reverse primers, and 7.5 μL SYBR Green Master Mix in a total volume of 15 μL, according to the manufacturer's instructions. The *TIP41* gene was used as an internal control.

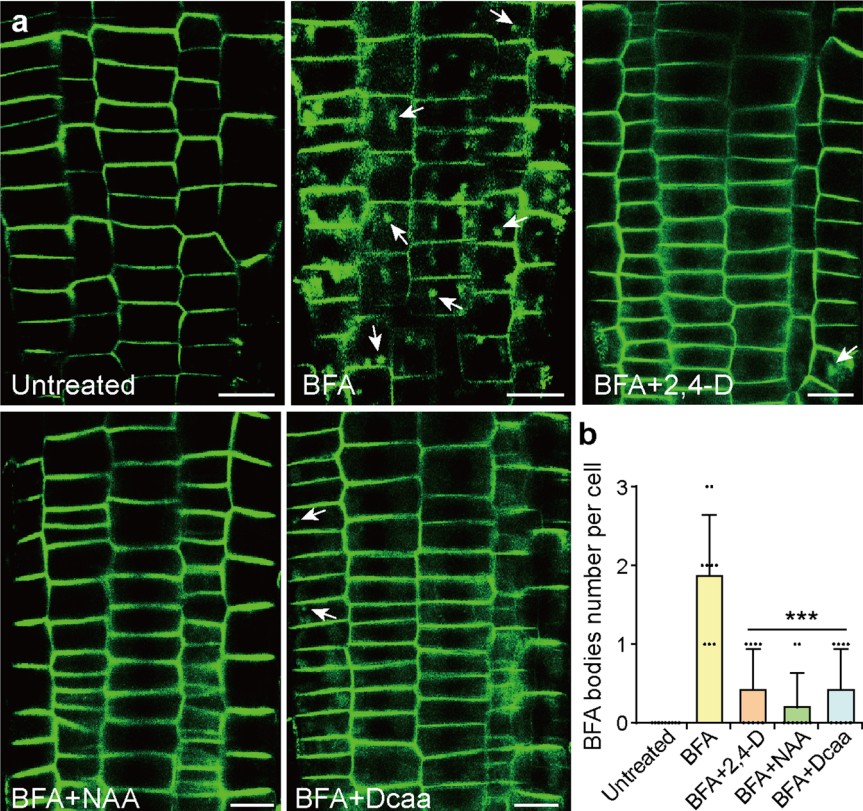

**Fig. 10 Dcaa inhibits endocytosis of PIN2 protein. a** Pre-treated 5-day-old *PIN2-GFP* seedlings with DMSO, 5 μM 2,4-D, 5 μM NAA, and 75 μM Dcaa for 30 min, then observed the intracellular localization of PIN2-GFP after adding 25 μM BFA co-treated with reagents mentioned above for 30 min. White arrows indicate BFA bodies. Bars = 25 μm. **b** Quantification of BFA bodies per cell in each treatment. Error bars represent the SD of the mean of 9 cells per treatment. ***$P < 0.001$ (Student's *t*-test, two-tailed, two-sample equal variance).

The relative transcription level was calculated by the $2^{-\Delta\Delta Ct}$ (Ct, cycle threshold) value. Primers used in the RT-qPCR analysis are listed in Supplementary Table 1.

**Arabidopsis root growth assay**. Sterilized seeds of Col, *tir1-1*, *tir1-1 afb1,2,3* were sown on MS solid medium containing 100 nM 2,4-D and 2.5 and 5 μM Dcaa, respectively, and cultured for 7 days in an incubator at 22 °C under a 16 h light/8 h dark cycle. Sterilized seeds of Col, *pin2-T*, and *aux1-T* were sown on MS solid medium containing 2.5 μM Dcaa, 50 nM 2,4-D, and 200 nM NAA, respectively, and cultured under the same conditions for 7 days. Then, photos were taken using a stereo microscope (Leica, Wetzlar, Germany), and root length was measured using ImageJ software.

**Molecular docking**. The 3D structure of Dcaa was constructed using Sybyl 6.9 (Iripos Inc., St. Louis. MO), and the molecule was then subjected to energy minimization at a gradient of 1.0 kcal/mol with a delta energy change of 0.05 cal/mol. The crystal structure (PDB: 2P1Q) of TIR1 was used as the receptor[49]. Other crystal structures of auxin receptor AFB1-5 were obtained by homologous modeling (https://swissmodel.expasy.org/)[64–67]. Mgltools 1.5.6 was used to prepare the ligand and receptors. Vina v1.2.3 was used to calculate the binding energies of ligands to the receptors. The best binding mode of each system was selected based on the binding energy.

**Auxin application assay at the Arabidopsis seedlings root tip**. Sterilized *DR5:GUS*, *pin2-T DR5:GUS*, *aux1-T DR5:GUS*, and *DR5rev:GFP* seeds were sown on MS solid medium and cultivated vertically for 5 days in an incubator under a 16 h light/8 h dark cycle at 22 °C. Subsequently, a strip of plastic wrap (approximately 2 cm in width) was placed onto a new MS medium to prevent the diffusion of applied auxin into the medium. Then, gently transfer the 5-day-old *DR5:GUS*, *pin2-T DR5:GUS*, *aux1-T DR5:GUS*, or *DR5rev:GFP* seedlings onto the new MS medium with only the root tips (approximately 0.2 mm) on the plastic wrap. A 1% agar gel strip containing 100 μM IAA, 50 μM 2,4-D, 100 μM NAA, or 1.5 mM Dcaa was placed onto the plastic wrap with just the upper edge on top of the root tips. Then the MS plates were incubated vertically for 13 h under the growth conditions described above. Subsequently, the *DR5:GUS*, *pin2-T DR5:GUS*, and *aux1-T DR5:GUS* seedlings were incubated in a GUS staining solution at 37 °C for 30 min. The GFP fluorescence and GUS staining signals were observed and recorded by a fluorescent or bright field microscope (Leica, Wetzlar, Germany)[68].

**BFA treatments**. Five-day-old *PIN2-GFP* seedlings were pre-treated with either 0.1% (v/v) DMSO, 5 μM NAA, 5 μM 2,4-D, or 75 μM Dcaa for 30 min and then co-treated with 25 μM BFA (Selleck Chemicals, Shanghai, China) or 0.1% (v/v) DMSO for 30 min. Confocal laser scanning microscopy (Leica TCS SP5, Leica, Wetzlar, Germany) was used to capture pictures. GFP was excited at 488 nm, and emission was detected at 500–550 nm. Images were processed with Adobe Photoshop CS6 and assembled in Adobe Illustrator CS4.

**Statistics and reproducibility**. Student's *t*-test was used to analyze the data. Error bars represent mean ± SD. Statistical significance and the number of biological repeats were described in the legend of each figure. All experimental results in the

manuscript, including fluorescence and GUS staining images have been pre-tested and assayed at least twice to ensure the accuracy of the experimental results. Representative experimental results were selected and presented in the manuscript. Statistical significance was set at $P < 0.05$, $P < 0.01$, and $P < 0.001$.

**Reporting summary**. Further information on research design is available in the Nature Portfolio Reporting Summary linked to this article.

## Data availability

All relevant data can be found within the manuscript and its supporting materials. The numerical source data for charts and graphs in the figures can be found in Supplementary Data 1. All other data are available from the corresponding author upon reasonable request.

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

## Acknowledgements

We thank M. Grebe for providing *aux1-T* and *pin2-T* seeds, J. Friml for *DR5rev:GFP* seeds, B. Scheres for *DR5:GUS* and *ProPIN2:PIN2-GFP* seeds, J. Pan and M. Estelle for *tir1-1* and *tir1-1 afb1,2,3* seeds. We thank Ruming Liu and Li Jiao for technical assistance in the use of the confocal. This work was supported by the National Natural Science Foundation of China (32070281, 32370271, 31870230, and 91417308 to S.M.).

## Author contributions

S.M., Q.B., X.Z., H.Z., W.X., and C.W. conceived the project and research plans; S.M. and C.T. designed the experiments; C.T. and S.L. performed the experiments; J.S, T.Z., and C.W. helped with the experiments; C.T. and S.M. analyzed the data; C.T. and S.M. wrote the article with the contributions of all the other authors.

## Competing interests

The authors declare no competing interests.
