## [Peer Review File · Communications Biology]

Reviewers' comments:

Reviewer #1 (Remarks to the Author):

In the article "3,4-Dichlorophenylacetic Acid - An Auxin Analogue Produces Beneficial Effects on Various Crops" by Chao Tan et al., the authors characterize a new auxin compound: 3,4-Dichlorophenylacetic Acid (Dcaa). The authors evaluated its effect in different species such as Arabidopsis, Tomato, Maize, Cabbage, Mung bean and Cucumber. Moreover, the authors also complemented these results with a more detailed characterization of its modulation of auxin signaling, signaling through TIR/AFB, regulation of gene expression, and transport. The results are certainly interesting, however, it is a bit difficult to understand the constant change of the auxin form the authors used as a comparison. Also, I have some doubts about the auxin transport assay (see comments below). Please find below more specific comments.

- Line 78: It is interesting that the authors mention that they screened 82 compounds out of 2000 chemicals synthesized. However, this is not mentioned in the results section.
- Line 94: Why phenylacetic acid and 2,4-Dichlorophenoxyacetic acid were not tested for the elongation of oat coleoptile segments? Interestingly, these two molecules are the most similar to Dcaa as mentioned in line 82.
- Line 96 and others: NAA or 1-NAA? In the introduction is mentioned as 1-NAA, then it is just NAA. Since the authors probably refer to the same compound please use the same abbreviation.
- Line 125: It is odd that the comparison is with a mix of IBA and NAA, not 2,4-D.
- Line 149: The previous point is even more relevant here, since the ck treatment has a strong negative effect in cabbage. The authors explain this due to an inhibitory effect. However, other than fresh and dry weight all the other parameters were not significant in ck, which is not very consistent with the explanation.
- Line 181: I think this needs more clarification. Foliar and root application was not done in the same species.
- Line 183 (Fig 2): Please add in the axis that the units are per hypocotyl.
- Line 271: It is a bit inconsistent that the authors change the auxin used as control depending on the experiment. Also as a general comment, it is also inconsistent that sometimes the units are ppm, others molarity.
- Line 300: correct Dcaa for Dcaa
- Line 314: I am not sure about this assay (Fig 8A-B), from the figure it is clear that auxin treatment results in DR5 upregulation. However, I think it is an extrapolation to conclude that it is due to auxin transport. Moreover, in some images, particularly from SFig6A, the gel strip is positioned over the root tip. Can the authors clarify/explain in more detail this assay?
- Line 328: Not sure about this statement. Perhaps the authors could change the term root tip for a more specific definition. In Fig8B there seems to be a clear difference between IAA and Dcaa. Moreover, the authors refer to the elongation zone with white arrows, but all that region is out of focus for untreated and Dcaa. Second, it is not clear how many roots the authors used for the quantification and the number of replicates.
- Line 286: 12-33 seedlings is a big difference. Also, that number is per replicate? How many replicates?
- Line 501: In the Supp Fig. 2 the authors mention 5ppm IBA+NAA, then in methods mention 1mg/L. Also, linked to previous comments, how do the authors define the IBA+NAA concentrations for the experiments? Seems to be that it was not standardized.
- Line 527: Please add reference for ImageJ.
- Time of GUS staining, 3h seems to be a short time considering that even GUS staining in control without auxin is saturated.

Reviewer #2 (Remarks to the Author):

The manuscript of Tan et al (COMMSBIO-23-1379-T) brings a set of results characterizing the effects of 3,4-Dichlorophenylacetic acid (here abbreviated Dcaa), the compound with a chemical structure related to auxin. Authors show that this compound improves the rooting of several crops, stimulates auxin-driven gene expression, and interferes with auxin transport and endocytosis in planta. While basic results on the molecular mechanism of action of this compound are from Arabidopsis, its potential utilization in agriculture is suggested based on the rooting assays in several crops. My overall impression of the work is that the selling point is not clear to me. Although the authors state that they present the result of the screen, it is not presented here, they just mention that this compound was selected based on the screen (non-described). 3,4-dichlorophenylacetic acid was already shown to be a very potent inhibitor of IAA influx in the past. In contrast, based on the experimentation that is not suitable for determining auxin transport it is concluded that it is rather auxin efflux than influx that is influenced. Also, it is not fully clear how big advantage for agriculture is to use 3,4-dichlorophenylacetic instead of 1-NAA or its derivatives. It is argued that actually less potent auxins like Dcaa are better thanks to their lower inhibitory interference with the growth of plant roots, which is true, but I am afraid that perhaps this is not enough to introduce this compound as the new auxin analog that is far better than those, which are used today. I have several specific comments.

Major comments:

- 1) At the end of the introduction and the beginning of the discussion, it is mentioned that in this study authors screened for new auxin-like compounds. However, this screen is not presented in this manuscript at all, there is just a brief mention that 2000 compounds were tested and 82 showed auxin-like activity (line 97), and that Dcaa "is the most promising". It is difficult to evaluate these statements because details of the screen are not presented and no reference is given.
- 2) The experiments on the elongation of oat coleoptiles, here I did not catch well the logic behind this setup. If authors want to compare the effectivity of Dcaa with auxin, I would expect an easy comparison with IAA and 1-NAA, and as a control, benzoic acid or tryptophan, or 2-NAA could be involved. Yes, here it is introduced to compare it with synthetic "auxins" used in agriculture. The sentence in line 105 "To analyze the auxin activity of Dcaa, we performed the classical oat coleoptile segments elongation assay and used NAA, IBA, 2-naphthoxyacetic acid, 1-naphthylacetamide and vitamin B1 (VB1) as control, which is widely used as plant growth regulators (Figure 1, A)" is awkward. What specifically is widely used as "plant growth regulators"? Figure 1C plot has y axis wrongly marked, there cannot be anything like "growth length", I am also afraid that 30 values are not enough for constructing violin plots, it would be good to show every individual value as a dot.
- 3) Concentration of compounds tested in this work are expressed here in three different forms, as molar, in ppm, and also in mg/L. This needs to be urgently unified, all of them should be shown as molar concentrations. In its current form, it is very hard to compare the efficiency of individual compounds.
- 4) In the introduction and/or discussion, it should be mentioned that 3,4-Dichlorophenylacetic acid (Imhoff et al., 2000 PMID: 10787051) and closely related compounds were already shown in the literature to interfere with auxin transport. Namely, CHPAA, which has besides blocking carrier-mediated influx (Lankova et al., 2010 PMID: 20595238) also auxin-like activity in higher concentrations (Parry et al., 2001 PMID: 11260496).
- 5) Figure 4 shows that Dcaa stimulates DR5::GUS in Arabidopsis roots. I am not surprised, considering that the related compound CHPAA, described firstly in Imhoff et al. 2000 (PMID:

10787051), showed a very clear upregulation of DR5-GUS (our unpublished results). However, I have two comments here. Firstly, the method of GUS staining is mentioned here to be performed the same as in Song et al 2019, but there, the method was used for embryos. And more importantly, it looks like that the signal that is induced in auxin-treated plants is shown here to be primarily in the root tip, but this is not the primary location, where the reaction to exogenously administered auxin should be scored if one would like to know whether the particular compound stimulates DR5-driven reporter. The applied hormone should primarily induce the signal in the surface layers of the root, as nicely shown in Simon et al. 2013 (PMID: 23914741). I wonder, how to interpret data presented in Figure 4?

6) Assays presented in Figure 8 and S6 are not in reality auxin transport assays, but they could be understood as indirectly showing the distribution of auxin signal within the root. It is entirely not known what auxin it is, it could easily be that Dcaa stimulates the production of the native auxin (IAA), which is transported. It could also be, what is more probable, that although authors tried to prevent diffusion of auxin or Dcaa into the medium these compounds in reality diffused through the medium. While GUS signal is brutally enhanced in the very tip, then there is a lack of the signal and further, it again goes up. In contrast, Dcaa is normal in the tip, then there is a strong drop in the signal and then it is again brutally enhanced. I would be very happy if the authors could comment on that, considering also that 3,4-Dichlorophenylacetic acid was shown to be interfering very potently with carrier-mediated auxin influx and much less with the efflux (Imhoff et al., 2000 PMID: 10787051). I do not feel that these assays could be conclusive enough for stating whether the particular compound is transported or not. For this, one would need dedicated radiolabel-based assays.

7) IBA potassium salt or 1-NAA sodium salt are both easier to dissolve in water, but their transport cannot be expected the same as IBA or 1-NAA, therefore I do not see the logic in comparing them with 3,4-Dichlorophenylacetic acid. Perhaps this logic is mostly only like comparing their agronomical usage, not mechanisms of transport of these molecules.

8) The technique of root irrigation is not described in methods, was it drop irrigation?

Minor comments:

1) All in-text references to images are unusual, like line 100 (Figure 1, A), instead Figure 1A. This is consistent throughout the text. Moreover, for Comm Biol the style should be rather "Fig. 1a"

2) Line 41 - transporters not transports

3) line 88 - *ex vitro* in italics

4) line 130 - "root genesis" is very unusual, the commonly used term is "root development"

Reviewer #3 (Remarks to the Author):

The manuscript described the characterization of a new auxin analog: the molecule 3,4-Dichlorophenylacetic Acid (Dcaa). The manuscript described the effect of this molecule on the growth of different crop species which is valuable for further uses in the field. It also demonstrates that Dcaa is an auxin signaling agonist similar to NAA for instance. The molecular effect of the Dcaa is elegantly shown using standard methodology and controls.

At the end of the introduction, the authors mentioned a screening of 82 chemicals. However, there is no detail of how the screening was performed or the criteria used to select Dcaa. Indeed, there is no mention of the output of the screening. However, the discussion starts with the sentence "In this study, we screened for new auxin-like compounds that can be used in broad species of crops." I recommend modifying the text and declaring only what is reported with results.

The manuscript is well-organized and clear for reading. The figures are self-explanatory.

Reviewer #1

1. Line 78: It is interesting that the authors mention that they screened 82 compounds out of 2000 chemicals synthesized. However, this is not mention in the results section.

Reply: In the revised version, we have added one subtitle “Design and screen of Dcaa” in the result part and have added the screening flow chart in the new Figure 1.

2. Line 94: Why phenylacetic acid and 2,4-Dichlorophenoxyacetic acid were not tested for the elongation of oat coleoptile segments? Interestingly, these two molecules are the most similar to Dcaa as mention in line 82.

Reply: Our original purpose is to screen an efficient and non-toxic plant growth regulator that has potential use in crops. In agricultural production, 2,4-D is mainly used as a herbicide and sometimes used for fruit setting, but is rarely used for rooting (which is prone to drug damage), whereas the mixture of indole butyric acid and naphthylacetic acid is a well-known and highly accepted rooting product on the market. Therefore, we mainly used potassium indole butyrate and sodium naphthylacetate but not PAA and 2,4-D as controls.

In the revised version, we compared the activity of PAA and Dcaa in inducing the expression of the auxin-responsive reporter *DR5rev:GFP* through Western blot experiments (see Supplementary Figure 1).

3. Line 96 and others: NAA or 1-NAA? In the introduction is mention as 1-NAA, Then is just NAA. Since the authors probably refers to the same compound please use the same abbreviation.

Reply: Thanks. In the revised version, we used the same abbreviation for NAA.

4. Line 125: It is odd the comparison is with a mix of IBA and NAA, not 2,4-D.

Reply: As described above, in agricultural production, 2,4-D is mainly used as a herbicide and sometimes used for fruit setting, but is rarely used for rooting (which is prone to drug damage). The mixture of indole butyric acid and naphthylacetic acid is well-known in the market and has the highest acceptance of rooting, making it the most suitable control agent for rooting assays. For these reasons, we did not compare Dcaa with 2,4-D.

5. Line 149: The previous point is even more relevant here, since the ck treatment has a strong negative effect in cabbage. The authors explain this due to inhibitory effect. However, other than fresh and dry weight all the other parameters were not significant in ck, which is not very consistent with the explanation.

Reply: Previous study has been reported that exogenous application of auxins usually produces an inhibitory effect on root elongation (Evans *et al.*, 1994; Simon *et al.*, 2013). According to our results, the fresh and dry weights of the ck treated group were significantly lower than those of the untreated group. However, the agronomic traits such as plant height and root length were lower than those of the untreated group, and there was no statistically significant difference from our limited statistical results. This may be due to the inevitable differences between individuals in physiological experiments, resulting in a large degree of dispersion among data. However, it does not affect our experimental conclusion. Dcaa, as a plant growth regulator, can indeed promote the development of cabbage roots.

6. I think this needs more clarification. Foliar and root application was not done in the same species.

Reply: A great question. We know that different plants or different organs of the same plant have different sensitivities to auxin. Based on our extensive field assays on different crops, this is indeed the case. This is also why we use root irrigation on crops such as cucumber and maize, while spraying leaves on tomatoes. In the preliminary research, we conducted experiments on tomato seedlings through root irrigation. It was found that tomato roots are very sensitive to auxin. Therefore, we chose the method of foliar spraying.

7. Please add in the axe that the units are per hypocotyl.

Reply: Revised accordingly.

8. It is a bit inconsistent that the authors change the auxin used as control depending on the experiment. Also as a general comment, it is also inconsistent that sometimes the units are ppm, others molarity.

Reply: In the first half of the article, we conducted a large number of crop assays. We selected auxin analogues widely used as plant growth regulators in the market as the control. Therefore, we have unified the unit as mg/L (ppm). To verify the auxin activity of Dcaa on Arabidopsis, we used NAA, 2,4-D, and IAA as controls. The unit has been unified as molar concentration.

9. Line 300: correct Dcaa for Dcaa.

Reply: Revised accordingly.

10. I am not sure about this assay (Fig 8 A-B), from the figure it is clear that auxin treatment results in DR5 upregulation. However, I think it is an extrapolation to conclude that it is due to auxin transport. Moreover, in some images, particularly from SFig6A, the gel strip is position over the root tip. Can the authors clarify/explain in

more detail this assay?

Reply: We have replaced the original SFig6 with the new SFig7.

We have also done the transport assay in the *aux1-T* and *pin2-T* mutants (see new Fig. 9a-c), and the results clearly show that in *pin2-T* only 2,4-D treatment enhanced the GUS signal in the root differentiation zone, whereas in *aux1-T* only application of NAA and Dcaa at the root tip increased the GUS signal in the root differentiation zone. These results suggest that Dcaa needs efflux but not influx for intercellular transport.

We also added the details for the auxin basipetal transport assay in the “material and methods” part.

11. Not sure about this statement. Perhaps the authors could change the term root tip for a more specific definition. In Fig8B there seems to be a clear difference between IAA and Dcaa. Moreover, the authors refer to the elongation zone with white arrows, but all that region is out of focus for untreated and Dcaa. Second, it is not clear how many roots the authors used for the quantification and the number of replicates.

Reply: In the revised manuscript, we analyzed the basipetal transport of Dcaa in *aux1-T* and *pin2-T* mutants that contained the *DR5:GUS* reporter. The original Fig8 (a-c) has been the current SFig7. Indeed, it can be seen from the graph that there are some differences between the Dcaa and IAA treatment groups. The fluorescence intensity of DR5rev:GFP after IAA treatment was significantly stronger than the Dcaa treated group. There are also some differences in the distribution of fluorescence signals in the roots. The IAA treated roots seems to have fluorescence signals in the meristem transition zone. However, we only detected fluorescence signals in the elongation regions of the Dcaa treated roots. As mentioned earlier, these results aim to demonstrate that Dcaa, like IAA, can be transported. These data only represent the results of one experiment with 5-11 seedlings in each group. At the same time, to ensure the reliability of the data, we repeated it three times and selected one representative data for display.

12. 12-33 seedlings is a big difference. Also, that number is per replicate? How many replicates?

Reply: Due to the poor germination vitality of these seeds, we selected Arabidopsis with consistent growth for measurement and statistics, so there may be differences in quantity. To ensure the reliability of the data, we repeated it three times and selected one representative data for display.

13. In the Supp Fig. 2 the authors mention 5ppm IBA+NAA, then in methods mention 1 mg/L. Also, linked to previous comments, how the authors define the IBA+NAA concentrations for the experiments? Seems to be that it was not standardized.

Reply: We apologize for writing 1 ppm as 5 ppm in the supplementary data. It has now been corrected. In the IBA + NAA mixed solution the final concentration of IBA was 1 ppm, the final concentration of NAA was also 1 ppm.

14. Please add reference for ImageJ.

Reply: We have added it. Collins TJ. (2007) ImageJ for microscopy. Biotechniques 43: 25-30.

15. Time of GUS staining, 3h seems to be a short time considering that even GUS staining in control without auxin is saturated.

Reply: Thanks for the reviewer's suggestion. Based on our experience, the light blue color in the meristem and elongation zones of the control root tip is a floating color, not a real GUS signal.

Reviewer #2

1. At the end of the introduction and the beginning of the discussion, it is mentioned that in this study authors screened for new auxin-like compounds. However, this screen is not presented in this manuscript at all, there is just a brief mention that 2000 compounds were tested and 82 showed auxin-like activity (line 97), and that Dcaa “is the most promising”. It is difficult to evaluate these statements because details of the screen are not presented and no reference is given.

Reply: In the revised version, we have added one subtitle “Design and screen of Dcaa” in the result part and have added the screening flow chart in the new Figure 1.

2. The experiments on the elongation of oat coleoptiles, here I did not catch well the logic behind this setup. If authors want to compare the effectivity of Dcaa with auxin, I would expect an easy comparison with IAA and 1-NAA, and as a control, benzoic acid or tryptophan, or 2-NAA could be involved. Yes, here it is introduced to compare it with synthetic “auxins” used in agriculture. The sentence in line 105 “To analyze the auxin activity of Dcaa, we performed the classical oat coleoptile segments elongation assay and used NAA, IBA, 2-naphthoxyacetic acid, 1-naphthylacetamide and vitamin B1 (VB1) as control, which is widely used as plant growth regulators (Figure 1, A)” is awkward. What specifically is widely used as “plant growth regulators”? Figure 1C plot has y axis wrongly marked, there cannot be anything like “growth length”, I am also afraid that 30 values are not enough for constructing violin plots, it would be good to show every individual value as a dot.

Reply: We chose these chemicals as controls because they can promote plant root development in agricultural production and are also common plant growth regulators in market sales. Therefore, we hope to evaluate the effectiveness of Dcaa in promoting the growth of oat coleoptiles by comparing with these substances. We also selected common auxin analogues such as NAA and IBA for comparison. Due to the similarity in chemical structure between Dcaa and phenylacetic acid (PAA), we supplemented the WB assay to compare the activity of these two substances. We have deleted “which is widely used as plant growth regulators”. We have changed the y-axis to “Elongation of oat coleoptile segments (mm)”. We have changed the violin plot to column graphs displaying individual value.

3. Concentration of compounds tested in this work are expressed here in three different forms, as molar, in ppm, and also in mg/L. This needs to be urgently unified, all of them should be shown as molar concentrations. In its current form, it is very hard to compare the efficiency of individual compounds.

Reply: In the first half of the article, we conducted a large number of crop assays. We selected auxin analogues widely used as plant growth regulators in the market as the control. Therefore, we have unified the unit as mg/L (ppm). To verify the auxin

activity of Dcaa on Arabidopsis, we used NAA, 2,4-D, and IAA as controls. The unit has been unified as molar concentration.

4. In the introduction and/or discussion, it should be mentioned that 3,4-Dichlorophenylacetic acid (Imhoff et al., 2000 PMID: 10787051) and closely related compounds were already shown in the literature to interfere with auxin transport. Namely, CHPAA, which has besides blocking carrier-mediated influx (Lankova et al., 2010 PMID: 20595238) also auxin-like activity in higher concentrations (Parry et al., 2001 PMID: 11260496).

Reply: We have added these references in the Discussion section.

5. Figure 4 shows that Dcaa stimulates DR5::GUS in Arabidopsis roots. I am not surprised, considering that the related compound CHPAA, described firstly in Imhoff et al. 2000 (PMID: 10787051), showed a very clear upregulation of DR5-GUS (our unpublished results). However, I have two comments here. Firstly, the method of GUS staining is mentioned here to be performed the same as in Song et al 2019, but there, the method was used for embryos. And more importantly, it looks like that the signal that is induced in auxin-treated plants is shown here to be primarily in the root tip, but this is not the primary location, where the reaction to exogenously administered auxin should be scored if one would like to know whether the particular compound stimulates DR5-driven reporter. The applied hormone should primarily induce the signal in the surface layers of the root, as nicely shown in Simon et al. 2013 (PMID: 23914741). I wonder, how to interpret data presented in Figure 4?

Reply: We have replaced “Song et al 2019” with “Wang et al 2021”.

We also observed GUS staining in the surface layers as displayed in the new Fig. 9a of Dcaa treatment. However, in Fig. 4, in order to show the GUS staining in the QC and columella cells we chose the images that focus on the QC, which showed clear enhance of GUS staining in these cells after Dcaa treatment. Furthermore, the GUS activity assay was done on the whole seedlings, which also showed significant increase of GUS activity after Dcaa treatment.

6. Assays presented in Figure 8 and S6 are not in reality auxin transport assays, but they could be understood as indirectly showing the distribution of auxin signal within the root. It is entirely not known what auxin it is, it could easily be that Dcaa stimulates the production of the native auxin (IAA), which is transported. It could also be, what is more probable, that although authors tried to prevent diffusion of auxin or Dcaa into the medium these compounds in reality diffused through the medium. While GUS signal is brutally enhanced in the very tip, then there is a lack of the signal and further, it again goes up. In contrast, Dcaa is normal in the tip, then there is a strong drop in the signal and then it is again brutally enhanced. I would be very happy if the authors could comment on that, considering also that 3,4-Dichlorophenylacetic acid was shown to be interfering very potently with

carrier-mediated auxin influx and much less with the efflux (Imhoff et al., 2000 PMID: 10787051). I do not feel that these assays could be conclusive enough for stating whether the particular compound is transported or not. For this, one would need dedicated radiolabel-based assays.

Reply: In the revised manuscript, we have also analyzed the transport of Dcaa in *aux1-T* and *pin2-T* mutants that contained the *DR5:GUS* reporter (see new Fig. 9a-c). The results clearly show that compared to the wild type, in *pin2-T* only 2,4-D (does not need efflux for transport) treatment enhanced the GUS signal in the root hair zone, whereas in *aux1-T*, only application of NAA (does not need influx) and Dcaa at the root tip increased the GUS signal in the root hair zone. These results on one hand suggest that the GUS signals in the root hair zone are induced by hormones transported from the root tip, on the other hand indicate that Dcaa needs efflux but not influx for intercellular transport.

As for the reviewer's doubt that the auxin in the agar gel strip may diffuse into the culture medium, we also thought of this. In addition to placing plastic film under the agar gel strip, we conducted untreated and treated groups on the same MS plate. If auxin permeates into the medium, the untreated group should be no different from the treated group.

7. IBA potassium salt or 1-NAA sodium salt are both easier to dissolve in water, but their transport cannot be expected the same as IBA or 1-NAA, therefore I do not see the logic in comparing them with 3,4-Dichlorophenylacetic acid. Perhaps this logic is mostly only like comparing their agronomical usage, not mechanisms of transport of these molecules.

Reply: Yes, that's true. IBA potassium salt or 1-NAA sodium salt are only used to compare their agronomical usage, not mechanisms of transport.

8. The technique of root irrigation is not described in methods, was it drop irrigation?

Reply: It is not drop irrigation. We used pot plants and just poured the same volume of solution into the soil as depicted in Supplementary Fig. 5a.

9. Minor comments:

- 1) All in-text references to images are unusual, like line 100 (Figure 1, A), instead Figure 1A. This is consistent throughout the text. Moreover, for Comm Biol the style should be rather "Fig. 1a"
- 2) Line 41 - transporters not transports
- 3) line 88 - *ex vitro* in italics
- 4) line 130 - "root genesis" is very unusual, the commonly used term is "root development"

Reply: We have revised them accordingly.

Reviewer #3

At the end of the introduction, the authors mentioned a screening of 82 chemicals. However, there is no detail of how the screening was performed or the criteria used to select Dcaa. Indeed, there is no mention of the output of the screening. However, the discussion starts with the sentence “In this study, we screened for new auxin-like compounds that can be used in broad species of crops.” I recommend modifying the text and declaring only what is reported with results.

The manuscript is well-organized and clear for reading. The figures are self-explanatory.

Reply: In the revised version, we carefully considered the comments of the reviewer and added our screening ideas and flowchart, as can be seen in results section under the subtitle “Design and screen of Dcaa” and in the new Fig. 1.

Reviewers' comments:

Reviewer #1 (Remarks to the Author):

I think that the revised manuscript by Tan et al. shows a significant improvement. However, I still have a few doubts that could be clarify.

L88: With the new Fig 1 is easier to follow the story and the origin of Dcaa, the data is still missing. In fact, in line 97-98 the authors state "Through pot experiments and field trials, we ultimately determined that Dcaa (III-7) is a potential and efficient PGR that can promote plant growth". However, no data is shown. Perhaps the data leading to this conclusion can be added as a supplementary figure.

L338: please change to less sensitive to...

L350: I am still not completely convinced about the statement "Dcaa can be transported basipetally in the root and requires the auxin efflux carrier protein". I still think the authors are extrapolating the results by using a reporter line for auxin signaling. However, I think it is a good result that can be shown if the statement is re-phrased. Also, please mention the number of replicates and seedlings per replicate.

SFig 6: For Dcaa, not sure that the white arrow is in the elongation zone. I will say no.

Reviewer #2 (Remarks to the Author):

The manuscript is now improved, but I still have several points that need to be addressed. Following are my comments to individual points from the rebuttal.

1. A brief description of the screen is now provided. I appreciate that authors openly inform that there actually were not all 2000 compounds screened for auxin-like activity, as it was mentioned in the previous version....., "we have designed and synthesized more than 2000 chemicals. From them, we have screened 82 compounds with auxin-like activity". On the other hand, now the end of the introduction does not fit what is presented in the description of the screen. I would kindly ask the authors to come back to this again and state how was the screen done. Was it like it is mentioned at the end of the introduction, i.e. that "we have designed and synthesized more than 2,000 chemicals. From them, we have screened out 82 compounds with auxin-like activities" or was it rather like it is now described in the first part of results, i.e. that "More than 2000 compounds were designed by the "me too" and "active substructure splicing" methods. Based on factors such as the availability of raw materials, difficulty in synthesis, compound stability and toxicity, and cost control, 82 compounds were screened out for further physiological activity tests (Fig. 1b)". These two statements tell very different stories, I would like to believe that the latter is true, but I am leaving this on authors. Last but not least, the screen is described briefly with five stages. However, it is still not possible to evaluate if Dcaa came out as the most potent because no details of the screen are provided. If these details would be published in some other paper, it would be great, or if they could be attached as supplementary data here, it would be even better.
2. Western blot analysis is not convincing and it is not properly described in materials and methods. Fig. 2C y axis cannot be "elongation", because elongation is the process, units (mm) would not fit to this. It is probably the length, right?

3. I appreciate that the authors tried to unify units, but still, there are ppm appearing 32x in the text. I did not follow why only because these compounds are widely used on the market one should use ppm. Yes, if this manuscript serves as "communication" to farmers, perhaps this would fit. However, I doubt if farmers would profit from western blot (S1), where units are ppm, here molar concentration would fit much better. Moreover, mg/L is not ppm, this could be mg/kg, but not mg/L.

4. OK!!

5. In both Fig. 9 and Fig 5 (not 4) the concentrations of auxins are so extremely high that I capitulate to conclude on these results, this is out of any range I can think about. I also did not understand why there is DR5 maximum missing in columella initials in pin2-T SALK line (which is not specified in methods, same like aux1-T). At least what I remember from older Sabatini's works, eir1 has DR5 signal there.

6. The statement that "2,4-D does not need efflux for transport" is misleading. 2,4-D is uptaken by carrier-mediated influx, but for the efflux carriers, it is an extremely bad substrate, it is also very badly diffusing out. I agree that 1-NAA does not need an influx carrier (authors incorrectly state only "influx"), it is a very lipophilic compound that is of course taken by influx carriers, but thanks to its lipophilicity its diffusion is massive, which is why it is uptaken freely. I still feel that it is an overestimation to conclude on the transport of Dcca from these experiments, also because the concentrations of auxins are extremely high. Perhaps one can improve this by not stating transport, but simply DR5 response after the administration of Dcca to the tip of the root.

We are very grateful for the reviewers' constructive comments and suggestions on the revision of our manuscript "3,4-Dichlorophenylacetic Acid Acts as An Auxin Analogue and Produces Beneficial Effects on Various Crops" (COMMSBIO-23-1379A). We have revised the manuscript according to the reviewer's detailed suggestions. The following are point-by-point responses to the reviewer's comments:

Response to Comments of Reviewer #1:

1. L88: With the new Fig 1 is easier to follow the story and the origin of Dcaa, the data is still missing. In fact, in line 97-98 the authors state "Through pot experiments and field trials, we ultimately determined that Dcaa (III-7) is a potential and efficient PGR that can promote plant growth". However, no data is shown. Perhaps the data leading to this conclusion can be added as a supplementary figure.

Respond: According to the reviewer's suggestion, we have added a "Supplementary Data 1", which summaries the results of relevant experiments we had conducted on some of the compounds.

2. L338: please change to less sensitive to...

Respond: Revised accordingly.

3. L350: I am still not completely convinced about the statement "Dcaa can be transported basipetally in the root and requires the auxin efflux carrier protein". I still think the authors are extrapolating the results by using a reporter line for auxin signaling. However, I think it is a good result that can be shown if the statement is re-phrased. Also, please mention the number of replicates and seedlings per replicate.

Respond: We have re-phrased the sentence to "The transport of Dcaa in the root requires the PIN2 auxin efflux carrier protein".

We also re-phrased the sentence "The transport of these substances was observed by GUS and GFP signals intensity" to "The transport of these substances can be indirectly indicated by the enhanced GUS and GFP signals intensity in the root elongation and maturation zone." and added "However, we cannot exclude that the enhanced expression of *DR5:GUS/DR5rev:GFP* was due to the sustained presence of the induced auxin signaling." at the end of the paragraph.

We have indicated the number of replicates and seedlings per replicate in the figure legends of Fig. 9.

SFig 6: For Dcaa, not sure that the white arrow is in the elongation zone. I will say no.

Respond: The reviewer is correct; the white arrow in Dcaa treatment should be indicating the transition zone. We have revised it.

Response to Comments of Reviewer #2:

1. A brief description of the screen is now provided. I appreciate that authors openly inform that there actually were not all 2000 compounds screened for auxin-like activity, as it was mentioned in the previous version....,we have designed and synthesized more than 2000 chemicals. From them, we have screened 82 compounds with auxin-like activity”. On the other hand, now the end of the introduction does not fit what is presented in the description of the screen. I would kindly ask the authors to come back to this again and state how was the screen done. Was it like it is mentioned at the end of the introduction, i.e. that “we have designed and synthesized more than 2,000 chemicals. From them, we have screened out 82 compounds with auxin-like activities” or was it rather like it is now described in the first part of results, i.e. that “More than 2000 compounds were designed by the “me too” and “active substructure splicing” methods. Based on factors such as the availability of raw materials, difficulty in synthesis, compound stability and toxicity, and cost control, 82 compounds were screened out for further physiological activity tests (Fig. 1b)”. These two statements tell very different stories, I would like to believe that the latter is true, but I am leaving this on authors. Last but not least, the screen is described briefly with five stages. However, it is still not possible to evaluate if Dcaa came out as the most potent because no details of the screen are provided. If these details would be published in some other paper, it would be great, or if they could be attached as supplementary data here, it would be even better.

Respond: Yes, the latter is true. We have changed the end of the introduction to “Based on the molecular structures of those existing auxinic compounds, we have designed more than 2,000 chemicals. From them, 82 compounds were screened out for further physiological activity tests, among which the No. 066 compound (3,4-dichlorophenylacetic acid, Dcaa) is the most promising.” . We also have added a table in the supplementary data, which summaries the results of the relevant experiments we had conducted on some of the compounds.

2. Western blot analysis is not convincing and it is not properly described in materials and methods.

Fig. 2C y axis cannot be “elongation”, because elongation is the process, units (mm) would not fit to this. It is probably the length, right?

Respond: We have redone the Western blot; please see the new supplementary Fig. 1. We have revised the description of the Western blot experiment.

We have changed the “Elongation of oat coleoptile segments” to “Elongation length of oat coleoptile segments”.

3. I appreciate that the authors tried to unify units, but still, there are ppm appearing 32x in the text. I did not follow why only because these compounds are widely used on the market one should use ppm. Yes, if this manuscript serves as “communication“ to farmers, perhaps this would fit. However, I doubt if farmers would profit from western blot (S1), where units are ppm, here molar concentration would fit much better. Moreover, mg/L is not ppm, this could be mg/kg, but not mg/L.

Respond: This article was completed in collaboration with a company that produces plant growth regulators, the pot experiments and field trials were performed by the company's biological activity testing personnel and they use to using "ppm" as the unit. Therefore, some assay results are presented in ppm. We have changed the "ppm" to molar concentration in the results of Western blot. Yes, the reviewer is correct, ppm is mg/kg.

4. OK!!

5. In both Fig. 9 and Fig 5 (not 4) the concentrations of auxins are so extremely high that I capitulate to conclude on these results, this is out of any range I can think about. I also did not understand why there is DR5 maximum missing in columella initials in *pin2-T* SALK line (which is not specified in methods, same like *aux1-T*). At least what I remember from older Sabatini's works, *eir1* has DR5 signal there.

Respond: In Fig. 5, we did short-term (2 h) auxin treatment in liquid MS, to ensure the effectiveness of auxins we used very high concentrations of auxins. In Fig. 9, because auxins were applied in agar gel strips for short time (13 h), we therefore also used very high concentrations of auxins. Usually, both short-term and exogenous application of auxin use high concentrations than long-term treatment on the MS medium. For example, in Ikeda et al. (Nat Cell Biol. 2009, 11(6):731-8) and Fischer et al. (Curr Biol. 2006, 16(21):2143-9), Sephadex G-50 beads that had been immersed in 1 mM NAA were applied to the root tip. In Li et al. (Front Plant Sci. 2023, 14:1192340), the hypocotyls of melon seedlings were submerged in 250 mg/L (approximately 1.43 mM) of IAA.

Yes, there is DR5 maximum in columella initials in *pin2-T* SALK line as shown in our previous publications (Liu et al., Biochem Biophys Res Commun. 2018, 507(1-4): 433-436; Wang et al., Int J Mol Sci. 2021, 22(1):437.). In Fig 9, in order to see clearly the differences between untreated and auxin treatments and between different auxins, we did the GUS staining in short time (30 min), in our hand at this short time of staining we observed clear signals in the epidermis of transition zone of the untreated roots, so we did not focus on the columella initials. We have added the staining time in the methods.

6. The statement that "2,4-D does not need efflux for transport" is misleading. 2,4-D is uptaken by carrier-mediated influx, but for the efflux carriers, it is an extremely bad substrate, it is also very badly diffusing out. I agree that 1-NAA does not need an influx carrier (authors incorrectly state only "influx"), it is a very lipophilic compound that is of course taken by influx carriers, but thanks to its lipophilicity its diffusion is massive, which is why it is uptaken freely. I still feel that it is an overestimation to conclude on the transport of Dcca from these experiments, also because the concentrations of auxins are extremely high. Perhaps one can improve this by not stating transport, but simply DR5 response after the administration of Dcca to the tip of the root.

Respond: Thanks for the information about the transport of 2,4-D, we have revised the statement in line 367 to “whereas 2,4-D requires influx carriers but is hardly exported by efflux carriers”. We have revised “influx” to “influx carriers”. We also revised line 481 and 482 to “whereas 2,4-D depends on influx carriers to enter the cell but can hardly be transported out of the cell by either diffusion or efflux carrier-mediated active transport”.

We have revised line 357 to “The transport of these substances can be indirectly indicated by the enhanced GUS and GFP signals intensity in the root elongation and maturation zone”. We also have added one sentence at the end of this paragraph as the following: However, we cannot exclude that the enhanced expression of *DR5:GUS/DR5rev:GFP* was due to the sustained presence of the induced auxin signaling.

Reviewers' comments:

Reviewer #1 (Remarks to the Author):

In the revised version of the manuscript "3,4-Dichlorophenylacetic Acid Acts as An Auxin Analogue and Produces Beneficial Effects on Various Crops" by Tan et al. most of the comments made before were properly addressed. However, I still have a conflict with Figure 9. For instance, the authors can't say that they perform "an auxin transport assay" as mentioned in line 357. I have to insist that with this experiment the authors make a very interesting link with PIN2, however, this is not an auxin transport assay. Moreover, in the methods, I also recommend to change the title "Auxin basipetal transport assay in the Arabidopsis root" (line 639). If the assay was with DR5, this is an auxin signalling assay.

Reviewer #2 (Remarks to the Author):

The second revision of the manuscript of Tan et al (COMMSBIO-23-1379B) is a further improved version of the original contribution characterizing the effects of 3,4-Dichlorophenylacetic acid on the rooting of several crops, stimulation of auxin-driven gene expression, and interference with auxin transport and endocytosis in planta. In general, the manuscript is now significantly improved. Authors also provided further information on their initial screening. I am still in question on how big advantage for agriculture is to use 3,4-dichlorophenylacetic instead of 1-NAA or its derivatives, but perhaps authors are aware much more than me about the context.

I am still not happy with the western blot analysis that is made after 2 h treatment with PAA or Dcaa. Firstly, after 2 hours it is too early to detect any convincing increase on the protein level and if yes, this needs to be shown with some real quantification and biological repetitions. I would suggest authors either not to show this analysis and decrease the strength of their conclusion that Dcaa has higher auxin activity than PAA, or, as the alternative, to include western blot analysis in time scale from 2 h onward (2, 4, 12, 24, or something similar). In general, I found the 2h treatments, which are used in this work, as too short to robustly show auxin-stimulated expression of marker proteins. Same, 2 h auxin treatments of DR5rev:GFP are not convincing (S6), also considering the number of roots analyzed and the absence of biological replicates.

We are pleased to re-submit our revised manuscript “3,4-Dichlorophenylacetic Acid Acts as An Auxin Analogue and Produces Beneficial Effects on Various Crops” for publication in *Communications Biology*.

Thank you very much for making a critical assessment on the second version of our manuscript. We are glad to hear the affirmation of our previous manuscript from two reviewers. However, they also raised two issues that need to be resolved and answered. We have now completed a thorough revision as per recommendations of the reviewers. Our point-by-point response to reviewers’ comments are listed below.

Reviewer 1: I still have a conflict with Figure 9. For instance, the authors can’t say that they perform “an auxin transport assay” as mentioned in line 357. I have to insist that with this experiment the authors make a very interesting link with PIN2, however, this is not an auxin transport assay. Moreover, in the methods, I also recommend to change the title “Auxin basipetal transport assay in the Arabidopsis root” (line 639). If the assay was with DR5, this is an auxin signalling assay.

According to the reviewer's comments, we have rephrased the following:

- (1) In the abstract, we have replaced “Dcaa may be transported basipetally in plants and requires PIN auxin efflux carrier” with “Application of Dcaa at the root tip of the *DR5:GUS* auxin-responsive reporter can induce *GUS* expression at the root hair zone, which requires the PIN2 auxin efflux carrier”.
- (2) We have replace the subtitle in Line 351 “The transport of Dcaa in the root requires the PIN2 auxin efflux carrier protein” with “Roots of *pin2-T* but not *aux1-T* mutants are hypersensitive to Dcaa treatment”.
- (3) Line 354 “we carried out an auxin transport assay” was changed to “we carried out an auxin application assay”.
- (4) Line 375-377 “To determine whether Dcaa requires the auxin influx/efflux carriers for intercellular transport, we first analyzed the basipetal transport of Dcaa in *aux1-T* and *pin2-T* mutants containing the *DR5:GUS* reporter” was changed to “To determine the effect of Dcaa on auxin transport mutants, we first analyzed the expression of the *DR5:GUS* reporter in *aux1-T* and *pin2-T* mutant roots after application of Dcaa at the root tip”.
- (5) The title of Fig 9 was changed to “Roots of *pin2-T* but not *aux1-T* mutants are hypersensitive to Dcaa treatment.” and the legends of a-c was changed to “GUS signals in the primary root of *DR5:GUS*, *pin2-T DR5:GUS*, and *aux1-T DR5:GUS* seedlings 13 h after agar gel strips containing 0 (Mock), 100 μ M IAA, 50 μ M 2,4-D, 100 μ M NAA, or 1.5 mM Dcaa were placed below the root tip”.
- (6) In the methods section the title “Auxin basipetal transport assay in the Arabidopsis root “ was changed to “Auxin application assay at the Arabidopsis seedlings root tip”.

Reviewer 2: the western blot analysis that is made after 2 h treatment with PAA or Dcaa. Firstly, after 2 hours it is too early to detect any convincing increase on the protein level and if yes, this needs to be shown with some real quantification and biological

repetitions. I would suggest authors either not to show this analysis and decrease the strength of their conclusion that Dcaa has higher auxin activity than PAA, or, as the alternative, to include western blot analysis in time scale from 2 h onward (2, 4, 12, 24, or something similar). In general, I found the 2h treatments, which are used in this work, as too short to robustly show auxin-stimulated expression of marker proteins. Same, 2 h auxin treatments of DR5rev:GFP are not convincing (S6), also considering the number of roots analyzed and the absence of biological replicates.

Thanks for the reviewer's comments. We have removed the western blot assay (Supplementary Fig.1) as suggested.

In Fig S6, we conducted a 2 h auxin treatment on *DR5rev:GFP*, which the reviewers thought may be too short and we lacked biological replicates. Because we had performed this auxin application assay on the *DR5:GUS* reporter and verified GUS expression by both histochemical staining and GUS activity assay (Fig. 5). We therefore only conducted one experiment repeat on *DR5rev:GFP* and put the results in the Supplementary Fig.

As mentioned above, the manuscript has been revised according to the comments of the reviewers. We appreciate your help with the manuscript. Finally, I would like to thank reviewers for their careful and patient reviews which contribute greatly to improving our manuscript.